# DeCaFlow: A Deconfounding Causal Generative Model

**Alejandro Almodóvar**[*1]     **Adrián Javaloy**[*2]     **Juan Parras**[1]     **Santiago Zazo**[1]     **Isabel Valera**[3,4]

[1]Information Processing and Telecommunications Center, Universidad Politécnica de Madrid, Madrid, Spain
[2]School of Informatics, University of Edinburgh, Edinburgh, United Kingdom
[3]Department of Computer Science, Saarland University, Saarbrücken, Germany
[4]Max Planck Institute for Software Systems, Saarbrücken, Germany

## Abstract

We introduce DeCaFlow, a deconfounding causal generative model. In stark contrast to prior works, DeCaFlow requires training once per dataset with observational data and the causal graph, and enables accurate causal inference on continuous variables under the presence of hidden confounders. We extend previous theoretical results to show that a single instance of DeCaFlow provides correct estimates for all causal queries identifiable with do-calculus, leveraging proxy variables when do-calculus alone is insufficient. Moreover, we extend these results to counterfactual queries as well. Our empirical results on datasets such as Ecoli70—with 3 independent hidden confounders, tens of observed variables and hundreds of causal queries—show that DeCaFlow outperforms existing approaches, while demonstrating its out-of-the-box applicability to any given causal graph.

## 1 INTRODUCTION

Causal inference concerns how changes in one variable affect others, which is crucial to evaluate the effects of interventions in real-world applications [14, 64, 74]. Often, however, empirical trials are infeasible due to ethical, financial, or practical constraints, and thus answering causal queries from observational data becomes essential. Unfortunately, this is a especially challenging task due to the presence of unmeasured hidden confounders [1, 19].

Our goal here is to enable practical and accurate causal inference on continuous variables under the presence of hidden confounders. To this end, we build on two key concepts: **i)** *causal generative models* (CGMs) [9, 25, 29, 55, 73], a class of generative models that can generate samples from

Figure 1: **DeCaFlow can be effortlessly applied to highly complex causal graphs**, as that of the Ecoli70 dataset [56], with multiple hidden confounders and dozens of variables. We dash hidden confounders, and highlight direct *hidden-confounded* effects as identifiable (and thus correctly estimated by DeCaFlow), or unidentifiable.

the observational, interventional and (sometimes) counterfactual distributions;[1] and **ii)** *proxy variables*, i.e., conditionally independent variables that yield information about the hidden confounders [41, 42, 42, 66]. Consequently, we introduce the DeCaFlow, a CGM which provides correct estimates of a broad class of interventional and counterfactual queries under hidden confounding and, in stark contrast with existing CGMs [66, 71, 72], it requires training once per dataset with only observational data and the causal graph.

We prove theoretically that *DeCaFlow correctly estimates interventional and counterfactual queries that are identifiable with do-calculus, leveraging proxy variables when do-calculus alone is insufficient.* Specifically, we first extend recent advances in proximal causal inference by Miao et al. [41] and Wang and Blei [66] to include counterfactual causal queries. Then, we integrate proximal-identifiability with do-calculus, expanding the number of identifiable queries of which DeCaFlow is shown to provide correct estimates.

---

*Equal contribution.

[1]We defer the reader to §E for a discussion on relevant works.

*Accepted for the 8th Workshop on Tractable Probabilistic Modeling at UAI (TPM 2025).*

As proof of the claimed flexibility, Fig. 1 shows the causal graph of the Ecoli70 dataset [56], comprising 43 observed variables and 3 hidden confounders, which DeCaFlow can effortlessly model despite the complex settings, and accurately recovers diverse causal effects after a single training process. Remarkably, green edges in the figure represent direct causal effects that DeCaFlow can estimate despite the presence of hidden confounders. We empirically validate all our claims on semi-synthetic and real-world experiments, demonstrating that DeCaFlow outperforms existing alternatives while being widely applicable out-of-the-box.

## 2 BACKGROUND

**Definition 1.** A *(confounded) Structural Causal Model (SCM)* is a triplet $\mathcal{M} := (\mathbf{f}, P_{\mathbf{u}}, P_{\mathbf{z}})$ describing a data-generating process over $D$ observed (endogenous) variables $\mathbf{x} := (\mathrm{x}_1, \mathrm{x}_2, \ldots, \mathrm{x}_D) \in \mathcal{X}$:

$$\mathrm{x}_i := f_i(\mathrm{pa}(i), \mathrm{u}_i, \mathbf{z}) \quad \text{for} \quad i = 1, 2, \ldots, D, \quad (1)$$
$$\text{with } \mathbf{u} := (\mathrm{u}_1, \mathrm{u}_2, \ldots, \mathrm{u}_D) \sim P_{\mathbf{u}}, \ \mathbf{z} \sim P_{\mathbf{z}},$$

where $f_i$ is the causal mechanism to compute $\mathrm{x}_i$ from its observed *causal parents*, $\mathrm{pa}(i)$, the i-th exogenous variable, $\mathrm{u}_i$, and the *hidden confounders*, $\mathbf{z} \in \mathcal{Z}$.

While we make the dependence on the hidden confounders explicit for all observed variables in Eq. 1, we assume w.l.o.g. that a subset of them may not be directly affected by the hidden confounders. Furthermore, given a SCM $\mathcal{M}$, we denote by $\mathcal{G}$ the *faithful* causal graph that it induces, representing a direct causal relationship between pairs of endogenous and hidden variables *only* if it exists.

**Definition 2.** A *causal query* $Q(\mathcal{M}) := p_{\mathcal{M}}(\mathrm{y} | \mathrm{do}(\mathrm{t}), \mathbf{c})$ is a distribution over $\mathrm{y} \in \mathbf{x}$ (the *outcome* variable), as a result of intervening upon the variable $\mathrm{t} \in \mathbf{x}$ (the *treatment* variable). Additionally, $Q(\mathcal{M})$ denotes an *interventional* or *counterfactual* query if the variable $\mathbf{c}$ is, respectively, the empty set or the vector of observed factual values, $\mathbf{x}^{\mathrm{f}}$.

We call a causal query *identifiable* if it can be expressed as a function of the observational distribution, $p_{\mathcal{M}}(\mathbf{x})$, and the causal graph $\mathcal{G}$ [47]. As a result, any SCM inducing the same graph and matching the observational distribution produces correct estimates of that causal query. Moreover, *any* identifiable query can be rewritten this way using a set of three rules, the *do-calculus* [46], yet in the presence of *hidden confounders* this may not be possible and we risk producing incorrect estimates due to unaccounted confounders.

**Causal normalizing flows (CNFs)** [25] are the basis of DeCaFlow, given their strong guarantees despite mild assumptions. Given a causal graph $\mathcal{G}$, a CNF $T_{\theta}$ is a masked autoregressive normalizing flow [44] built such that it defines an unconfounded SCM $\mathcal{M}_{\theta} = (T_{\theta}, P_{\mathbf{u}})$ inducing $\mathcal{G}$ *by design*.

As demonstrated by Javaloy et al. [25], CNFs are a remarkable family of CGMs as they not only form a parametric

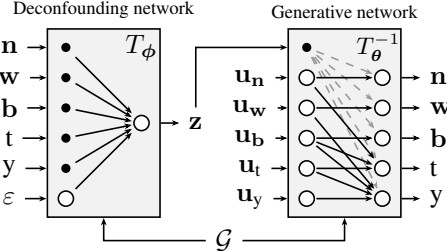

Figure 2: **Example of DeCaFlow architecture** for the causal graph $\mathcal{G}$ in Fig. 5 during training (Eq. 4). Circles represent input/output variables of the masked conditional normalizing flows, and black dots conditional inputs. $\varepsilon$ is a non-causal random variable needed to model $\mathbf{z}$ with $T_{\phi}$.

family of *identifiable SCMs*, but they can provably approximate the underlying SCM in the three rungs of Pearl's ladder of causation [47] simply by maximizing the observed joint evidence, i.e., $\max_{\theta} \log p_{\theta}(\mathbf{x})$. Furthermore, CNFs are also equipped with an *exact do-operator* for efficient sampling of any causal query, enabling their use for complex causal-inference tasks. Their main downside is the need to assume the absence of hidden confounders to guarantee the capabilities above, limiting their application.

## 3 DECONFOUNDING CAUSAL NORMALIZING FLOWS

Let $\mathcal{M}$ be an underlying confounded SCM $\mathcal{M}$, as in Def. 1, of which we have access to $N$ i.i.d. observations as well as to the faithful causal graph, $\mathcal{G}$. Our goal is to design and learn a CGM that can *accurately estimate as many causal queries from the original SCM as possible*, despite hidden confounding. In other words, we seek a substitute model of $\mathcal{M}$ to accurately perform causal inference.

**Assumptions.** We assume all variables to be continuous, and the SCM $\mathcal{M}$ to have $C^1$-diffeomorphic equations conditioned on $\mathbf{z}$, and to induce an acyclic causal graph $\mathcal{G}$.

We now present the *deconfounding causal normalizing flow*, or DeCaFlow for short, a CGM which extends CNFs [25] to account for hidden confounding while retaining all their theoretical properties. To achieve this, DeCaFlow follows an architecture akin to variational autoencoders [31] as shown in Fig. 2, i.e., DeCaFlow comprises two main components: **i)** *a generative network* that exploits structural constraints to faithfully model $\mathcal{M}$, given a substitute of $\mathbf{z}$; and **ii)** an *inference network* which approximates the *intractable* posterior distribution of $\mathbf{z}$ as modeled by the generative network, given the observed endogenous variables. In the following, we provide further details on both networks:

**Generative network.** We use CNFs [25] as our starting point, and adapt them to take the hidden confounders as conditional inputs by using conditional masked autoregressive normalizing flows [69]. The resulting model, $T_{\theta}$, is thus an

invertible transformation conditioned on $\mathbf{z}$, describing a data-generating process mapping a set of exogenous variables to endogenous ones and vice versa, i.e., $T_{\boldsymbol{\theta},\mathbf{z}}(\mathbf{x}) = \mathbf{u} \sim P_{\mathbf{u}}$ and $\mathbf{x} = T_{\boldsymbol{\theta},\mathbf{z}}^{-1}(\mathbf{u})$, where we further exploit the graph $\mathcal{G}$ to ensure that the generative process is faithful, i.e., that

$$p_{\boldsymbol{\theta}}(\mathbf{x} \mid \mathbf{z}) = \prod_{i=1}^{D} p_{\boldsymbol{\theta}}(\mathbf{x}_i \mid \mathrm{pa}(i), \mathbf{z}), \qquad (2)$$

similar to Def. 1 and, just as in that definition, *only the children of $\mathbf{z}$ are actually conditioned on $\mathbf{z}$* in Eq. 2.

**Deconfounding network.** To model the posterior of $\mathbf{z}$ given our observations as modeled by $T_{\boldsymbol{\theta}}$, i.e., the abduction step needed to compute counterfactuals [47], we use another masked autoregressive conditional normalizing flow [69], as it can approximate this distribution arbitrarily well. Once again, we exploit knowledge of $\mathcal{G}$ and mask the resulting network, $T_{\boldsymbol{\phi}}$, such that it models $\mathbf{z}$ using only the strictly necessary variables to ease its learning:

$$q_{\boldsymbol{\phi}}(\mathbf{z} \mid \mathbf{x}) = q_{\boldsymbol{\phi}}(\mathbf{z} \mid \mathrm{ch}(\mathbf{z}) \cup \mathrm{pa}(\mathrm{ch}(\mathbf{z}))). \qquad (3)$$

We provide in §C a more general version of Eq. 3, and an empirical validation on the choice of architecture in §B.2.

**Training process.** We jointly train both networks as typically done in deep latent-variable models, i.e., we maximize the evidence lower bound (ELBO) [31]:

$$\begin{aligned}
\mathcal{L}(\boldsymbol{\theta}, \boldsymbol{\phi}) &= \mathbb{E}_{q_{\boldsymbol{\phi}}}[\log p_{\boldsymbol{\theta}}(\mathbf{x}, \mathbf{z})] + \mathrm{H}(q_{\boldsymbol{\phi}}(\mathbf{z} \mid \mathbf{x})) \qquad (4) \\
&= \mathbb{E}_{q_{\boldsymbol{\phi}}}[\log p_{\boldsymbol{\theta}}(\mathbf{x} \mid \mathbf{z})] - \mathrm{KL}[q_{\boldsymbol{\phi}}(\mathbf{z} \mid \mathbf{x}) \| p(\mathbf{z})],
\end{aligned}$$

where $p(\mathbf{z})$ is the prior of $\mathbf{z}$, KL the Kullback-Leibler divergence [34], and H the differential entropy [32]. Optimizing the ELBO encourages that: **i)** the generative network explains the observations given samples from $q_{\boldsymbol{\phi}}$ (first term of Eq. 4); **ii)** the deconfounding network prevents allocating information exclusive of $\mathbf{x}$ in $\mathbf{z}$ (entropy term in Eq. 4); and **iii)** DeCaFlow matches the observation distribution, $p_{\mathcal{M}}(\mathbf{x})$, as all the theory relies on it. More specifically, the last point is encouraged since

$$\begin{aligned}
\max_{\boldsymbol{\phi}, \boldsymbol{\theta}} \mathcal{L}(\boldsymbol{\phi}, \boldsymbol{\theta}) = \min_{\boldsymbol{\phi}, \boldsymbol{\theta}} \ & \mathrm{KL}[p_{\mathcal{M}}(\mathbf{x}) \| p_{\boldsymbol{\theta}}(\mathbf{x})] \\
& + \mathrm{KL}[q_{\boldsymbol{\phi}}(\mathbf{z} \mid \mathbf{x}) \| p_{\boldsymbol{\theta}}(\mathbf{z} \mid \mathbf{x})]. \quad (5)
\end{aligned}$$

DeCaFlow is however susceptible to posterior collapse [67] as a result of using the ELBO, i.e., to the KL term in Eq. 4 precipitately vanishing, and the posterior hence equating the prior. Fortunately, we can leverage existing solutions and, e.g., employ annealing or KL balancing terms [63].

# 4 ESTIMATION OF CAUSAL QUERIES UNDER HIDDEN CONFOUNDING

By leveraging recent results in proximal-identifiability, we next show that DeCaFlow not only preserves the properties

of CNFs, but expand them. While we present here a short summary, all derivations can be found in §A.

## 4.1 INTERVENTIONAL QUERIES

First, we consider *hidden-confounded* interventional queries, i.e., queries of the form $Q(\mathcal{M}) = p_{\mathcal{M}}(\mathrm{y} \mid do(\mathrm{t}))$, where $\mathrm{y}, \mathrm{t} \in \mathrm{ch}(\mathbf{z})$ are any two children of the hidden confounder. We formalize the following proposition in §A.2:

**Proposition 4.1** (Informal). *A query $p_{\mathcal{M}}(\mathrm{y} \mid do(\mathrm{t}))$, where $\mathrm{y}, \mathrm{t} \in \mathrm{ch}(\mathbf{z})$ are two different children of $\mathbf{z}$, is identifiable if there exists a (potentially empty) subset of variables $\mathbf{b} \subset \mathbf{x} \setminus \{\mathrm{t}, \mathrm{y}\}$, and two proxies $\mathbf{n}, \mathbf{w} \in \mathbf{x} \setminus \{\mathrm{t}, \mathrm{y}, \mathbf{b}\}$ such that:*

1. $(\mathbf{b}, \mathbf{z})$ *forms a valid adjustment set,*
2. $\mathbf{w}$ *is a proxy variable given $\mathbf{b}$, i.e., $\mathbf{w} \perp\!\!\!\perp (\mathrm{t}, \mathbf{n}) \mid \mathbf{b}, \mathbf{z}$,*
3. $\mathbf{n}$ *is a null proxy variable given $\mathbf{b}$, i.e., $\mathrm{y} \perp\!\!\!\perp \mathbf{n} \mid \mathrm{t}, \mathbf{b}, \mathbf{z}$,*
4. *both $\mathbf{w}$ and $\mathbf{n}$ yield enough information about $\mathbf{z}$.*

Prop. 4.1 extends the results of Miao et al. [41] and Wang and Blei [66] to prove identifiable of queries under hidden confounding *even if treatment and outcome have observed parents in common*, rendering causal queries identifiable in the infinite-data regime by leveraging proxy information, thus complementing classical do-calculus [35]. Intuitively, $\mathbf{w}$ is used to build a function which "substitutes" the hidden confounder for that query, and $\mathbf{n}$ ensures that this substitute yields the correct estimate. Next, we expand the class of identifiable causal queries by introducing the queries identifiable with Prop. 4.1 as an additional base case for the recursive steps of do-calculus:

**Corollary 4.2.** *An interventional query is identifiable if, using do-calculus, it can be reduced to a combination of observational queries and identifiable interventional queries in the sense of Prop. 4.1.*

Similar to CNFs [25], we can readily interpret the generative network of DeCaFlow as a parametric confounded SCM (Def. 1) of the form $\mathcal{M}_{\boldsymbol{\theta}} := (T_{\boldsymbol{\theta}}^{-1}, P_{\mathbf{u}}, P_{\mathbf{z}})$. This SCM induces $\mathcal{G}$ by design, and since the family of normalizing flows are universal density approximators, $\mathcal{M}_{\boldsymbol{\theta}}$ can match the observational distribution $p_{\mathcal{M}}(\mathbf{x})$ given enough resources. We can then prove the following:

**Corollary 4.3.** *If DeCaFlow induces the same causal graph as $\mathcal{M}$ and $p_{\mathcal{M}}(\mathbf{x}) \stackrel{a.e.}{=} p_{\boldsymbol{\theta}}(\mathbf{x})$, then it correctly estimates any query identifiable in the sense of Cor. 4.2.*

## 4.2 COUNTERFACTUAL QUERIES

Next, we focus on queries $Q(\mathcal{M}) = p_{\mathcal{M}}(\mathrm{y}^{\mathrm{cf}} \mid do(\mathrm{t}^{\mathrm{cf}}), \mathbf{x}^{\mathrm{f}})$, where $\mathbf{x}^{\mathrm{f}}$ is an observed factual. Intuitively, this query represents *the distribution the outcome would have had, had we intervened on the treatment variable*. We demonstrate a one-to-one correspondence between proxy-identifiable interventional and counterfactual queries:

**Proposition 4.4** (Informal). *If a query $p(\mathrm{y} \mid do(\mathrm{t}))$ is identifiable in the sense of Prop. 4.1, then its counterfactual counterpart, $p(\mathrm{y}^{\mathrm{cf}} \mid do(\mathrm{t}^{\mathrm{cf}}), \mathbf{x}^{\mathrm{f}})$, is also identifiable.*

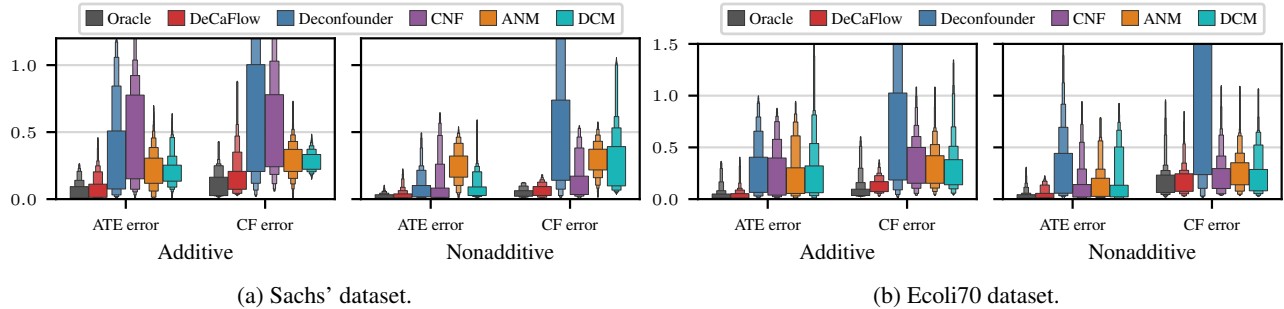

(a) Sachs' dataset.

(b) Ecoli70 dataset.

Figure 3: ATE and CF error boxenplots [21] of different CGMs on the (a) Sachs and (b) Ecoli70 datasets, aggregating over all identifiable direct effects after intervening on their 25th, 50th, and 75th percentiles over 5 random initializations.

The proof of Prop. 4.4 exploits the notion of twin SCM [5], which duplicates the structural equations for the factual and counterfactual worlds while sharing the exogenous variables, and the fact that Prop. A.2 (the formal version of Prop. 4.1) allows for queries with additional covariates as long as they do not form colliders, which is always the case with $\mathbf{x}^f$ in $p_{\mathcal{M}}(\mathbf{y}^{cf}|\operatorname{do}(\mathbf{t}^{cf}), \mathbf{x}^f)$. We can then follow the same derivations from the previous section to show that:

**Corollary 4.5.** *If DeCaFlow induces the same causal graph as $\mathcal{M}$ and $p_{\mathcal{M}}(\mathbf{x}) \stackrel{a.e.}{=} p_{\boldsymbol{\theta}}(\mathbf{x})$, then it correctly estimates any counterfactual query decomposable as a combination of (proxy-)identifiable queries using do-calculus.*

While the above results can look surprising at first, recall that we assume continuous endogenous variables and diffeomorphic causal generators (§3). Moreover, the correct estimation of counterfactual queries does not come without challenges: **i)** we need to accurately estimate $p_{\boldsymbol{\theta}}(\mathbf{z}|\mathbf{x})$, which is why it is crucial to correctly design and train $q_{\boldsymbol{\phi}}$; and **ii)** given $\mathbf{z}$ and $\mathbf{x}$, we need to accurately perform the abduction step. Fortunately, the latter step is trivialized using CNFs as generative networks, since they are bijective given $\mathbf{z}$.

## 5 EMPIRICAL EVALUATION

We empirically test DeCaFlow on two semi-synthetic datasets, showing that it accurately estimates interventional and counterfactual queries when the requirements of §4 are met. We provide all details and additional experiments in §B.

**Common evaluation.** We measure estimation quality using mean absolute error (MAE) of the average treatment effect (ATE) and the counterfactual samples. We also account for differences across observed variables by computing all errors over standardized variables.

**Baselines.** We consider three CGMs assuming causal sufficiency: CNFs [25]; ANMs [22]; and DCMs [9]; and the Deconfounder [65], which uses proxies similar to DeCaFlow, yet it needs to train once per outcome. We use as *oracle* a CNF [25] that *observes* the hidden confounders.

**Datasets.** We consider the Sachs [53] and Ecoli70 [56] datasets, and randomly generate non-linear SCMs inducing the same causal graph as the original dataset, see Figs. 1 and 16. We consider additive and nonadditive equations, measure the effect of interventions on the downstream nodes, and ensure when generating the SCM that the randomized effect of the hidden confounder is perceptible.

**Results.** We present a visualization of the results in Fig. 3, where we can observe that DeCaFlow consistently outperforms every considered CGM for both ATE and counterfactual errors, *staying on par with the oracle model*. Moreover, we appreciate a great difference in performance between DeCaFlow and CNFs, which corroborates the importance of the additions introduced by DeCaFlow, since a CNF is equivalent to DeCaFlow with $\mathbf{z}$ of size zero.

Moreover, Fig. 3b shows that DeCaFlow is able to closely match the performance of the oracle model, outperforming existing approaches. Remarkably, this experiment highlights every strength of DeCaFlow as it needs to: **i)** model several hidden confounders affecting different sets of variables; **ii)** correctly estimate all causal queries with proxy information; and **iii)** achieve the above in an agnostic manner, i.e., training the model out-of-the-box and *one single time*, despite the graph $\mathcal{G}$ having 43 observed variables.

## 6 CONCLUDING REMARKS

In this work, we have introduced DeCaFlow, a CGM that enables accurate estimation of interventional and counterfactual queries under hidden confounding. DeCaFlow expands on CNFs, preserving and expanding their theoretical properties, while offering several key advantages over prior approaches. Namely, DeCaFlow can be applied out-of-the-box to any given causal graph and, training once per dataset, it correctly estimates a broad class of (potentially hidden-confounded) interventional *and counterfactual* queries, in stark contrast with existing approaches.

Exciting future work includes the use of instrumental variables [20], as well as applying DeCaFlow to time-varying settings and to real-world problems such as decision support systems [54], or policy making [15], to name a few.

## ACKNOWLEDGMENTS

The authors would like to thank Luigi Gresele for useful discussions and comments which helped improving the quality of this work. AA has received the support of the *Synthema* and *Synthia* projects, funded by European Union's Horizon Europe, under grant agreement ID 101095530 and 101172872, respectively. In addition, *Programa Propio UPM* funded his stay at Saarland University. AJ received funding from the DFG grant 389792660 as part of TRR 248 – CPEC, and the *"UNREAL: a Unified Reasoning Layer for Trustworthy ML"* project (EP/Y023838/1) selected by the ERC and funded by UKRI EPSR. This work has also been supported by the project *"Society-Aware Machine Learning: The paradigm shift demanded by society to trust machine learning,"* funded by the European Union and led by IV (ERC-2021-STG, SAML, 101040177).

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

# A  CAUSAL IDENTIFIABILITY

## A.1  MODEL IDENTIFIABILITY

In this section, we briefly discuss the identifiability (in the sense of Xi and Bloem-Reddy [70]) of those variables that are indirectly confounded by $\mathbf{z}$ or not confounded at all, i.e., of those variables that are not children of any hidden confounder. As we discuss now, we can reduce our SCM (Def. 1) to a conditional one that only models these aforementioned variables, recovering the identifiability guarantees from Javaloy et al. [25].

To prove model identifiability, we resort to what we call the induced conditional SCM, which intuitively represents the original SCM where we restrict our view to a subset of variables, and assume the rest of the variables are given.

**Definition 3** (Induced conditional SCM). Given a SCM $\mathcal{M} = (\mathbf{f}, P_{\mathbf{u}}, P_{\mathbf{z}})$, and a subset of observed variables $\mathbf{x}' \subset \mathbf{x}$, we define the *induced conditional SCM of $\mathcal{M}$ given $\mathbf{x}'$*, denoted by $\mathcal{M}_{|\mathbf{x}'}$, to the SCM result of having observed $\mathbf{x}'$, and where causal generators and exogenous variables are restricted to only those associated with the unconditioned variables, i.e., $\mathbf{x} \setminus \mathbf{x}'$.

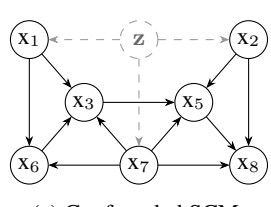

(a) Confounded SCM.

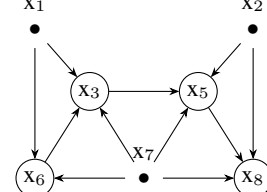

(b) Conditional unconfounded SCM.

Figure 4: Example of: **(a)** a confounded SCM $\mathcal{M}$; and **(b)** its induced conditional counterpart, $\mathcal{M}_{|\mathbf{x}'}$ where the children of the hidden confounder are observed and fixed, $\mathbf{x}' = \text{ch}(\mathbf{z}) = \{\mathbf{x}_1, \mathbf{x}_2, \mathbf{x}_7\}$. Note that $\mathcal{M}_{|\mathbf{x}'}$ does not exhibit hidden confounding.

We provide a visual depiction of this idea in Fig. 4. Using this definition, we can observe that, if we were to condition on the children of the hidden confounder, we would be left with a (conditional) *unconfounded SCM*, as the influence of the hidden confounder has been completely blocked by conditioning on its children. Now, if we have two models that perfectly match their marginal distributions, this means that they perfectly match their induced conditional SCM, no matter which value we observed for $\text{ch}(\mathbf{z})$, and we can thus leverage existing results from Javaloy et al. [25] for unconfounded SCMs. More specifically:

**Corollary A.1.** *Assume that we have two SCMs $\mathcal{M} := (\mathbf{f}, P_{\mathbf{u}}, P_{\mathbf{z}})$ and $\tilde{\mathcal{M}} := (\tilde{\mathbf{f}}, P_{\tilde{\mathbf{u}}}, P_{\tilde{\mathbf{z}}})$ that are Markov-equivalent—i.e., they induce the same causal graph—and which coincide in their marginal distributions, $p(\mathbf{x}) \overset{a.e.}{=} \tilde{p}(\mathbf{x})$. Then, both SCMs, restricted to every variable other than $\text{ch}(\mathbf{z})$, are equal up to an element-wise transformation of the exogenous distributions.*

*Proof.* The proof follows almost directly from [25, Theorem 1]. First, note that the two induced conditional SCMs are no longer influenced by $\mathbf{z}$ once that we have observed a specific realization of $\text{ch}(\mathbf{z})$, so that we can drop $\mathbf{z}$ from their structure, i.e., we can rewrite them instead as unconfounded SCMs, $\mathcal{M}_{|\text{ch}(\mathbf{z})} = (\mathbf{f}_{|\text{ch}(\mathbf{z})}, P_{\mathbf{u}|\text{ch}(\mathbf{z})})$ and $\tilde{\mathcal{M}}_{|\text{ch}(\mathbf{z})} = (\tilde{\mathbf{f}}_{|\text{ch}(\mathbf{z})}, P_{\tilde{\mathbf{u}}|\text{ch}(\mathbf{z})})$. To ease notation, let us call $\mathbf{x}^{\mathbf{c}} := \mathbf{x} \setminus \text{ch}(\mathbf{z})$ the variables that are not children of $\mathbf{z}$.

Next, note that for almost every realization of $\text{ch}(\mathbf{z})$, we have that $p(\mathbf{x}^{\mathbf{c}} | \text{ch}(\mathbf{z})) \overset{a.e.}{=} \tilde{p}(\mathbf{x}^{\mathbf{c}} | \text{ch}(\mathbf{z}))$ since $p(\mathbf{x}) \overset{a.e.}{=} \tilde{p}(\mathbf{x})$ by assumption and $p(\mathbf{x}) = p(\mathbf{x}^{\mathbf{c}} | \text{ch}(\mathbf{z})) p(\text{ch}(\mathbf{z}))$. As a result, for each realization of $\text{ch}(\mathbf{z})$ we can apply Theorem 1 of Javaloy et al. [25], which yields that the two induced conditional SCMs are equal up to an element-wise transformation of the exogenous distribution.

Finally, since the causal generators and exogenous distributions of the induced SCMs are, for almost every $\text{ch}(\mathbf{z})$, identical to their counterparts in the original SCMs (as we have just discarded those components associated with $\text{ch}(\mathbf{z})$), we get that, those elements in both SCMs associated with $\mathbf{x}^{\mathbf{c}}$, are identical up to said (possibly $\text{ch}(\mathbf{z})$-dependent) component-wise transformation. $\qquad\square$

## A.2 QUERY IDENTIFIABILITY

We now prove the identifiability of the causal queries considered in the main text.

To this end, one key property that we will use in the following is that of completeness (as, e.g., in the work of Wang and Blei [66]). Intuitively, we say that a random variable $\mathbf{z}$ is complete given another random variable $\mathbf{n}$ if "any infinitesimal change in $\mathbf{z}$ is accompanied by variability in $\mathbf{n}$" [42], yielding enough information to recover the posterior distribution of $\mathbf{z}$. This concept is similar in spirit to that of variability in the case of discrete random variables [43]. In practice, completeness is more likely to be achieved the more proxies we measure [3].

**Definition 4** (Completeness). We say that a random variable $\mathbf{z}$ is complete given $\mathbf{n}$ for almost all $\mathbf{c}$ if, for any square-integrable function $g(\cdot)$ and almost all $\mathbf{c}$, $\int g(\mathbf{z}, \mathbf{c}) p(\mathbf{z}|\mathbf{c}, \mathbf{n}) \, d\mathbf{z} = 0$ for almost all $\mathbf{n}$, if and only if $g(\mathbf{z}, \mathbf{c}) = 0$ for almost all $\mathbf{z}$.

The following proposition (informally simplified in Prop. 4.1) is a generalization of the results previously presented by Miao et al. [41] and Wang and Blei [66], where we include an additional covariate $\mathbf{c}$ to the causal query, and make no implicit assumptions on the causal graph allowing, e.g., for the treatment and outcome variables to hame some observed parents in common. However, note that $\mathbf{c}$ cannot be a collider (e.g., forming a subgraph of the form $\mathbf{n} \to \mathbf{c} \leftarrow \mathrm{y}$). Otherwise, conditioning on $\mathbf{c}$ would make independent variables dependent (in the example, y and $\mathbf{n}$), and the causal effect of t on y would not be identifiable.

**Proposition A.2** (Query identifiability). *Given two SCMs $\mathcal{M} := (\mathbf{f}, P_{\mathbf{u}}, P_{\mathbf{z}})$ and $\tilde{\mathcal{M}} := (\tilde{\mathbf{f}}, P_{\tilde{\mathbf{u}}}, P_{\tilde{\mathbf{z}}})$, assume that they are Markov-equivalent—i.e., they induce the same causal graph—and which coincide in their marginal distributions, $p(\mathbf{x}) \overset{a.e.}{=} \tilde{p}(\mathbf{x})$. Then, they compute the same causal query, $p(\mathrm{y}|\, do(\mathrm{t}), \mathbf{c}) = \tilde{p}(\mathrm{y}|\, do(\mathrm{t}), \mathbf{c})$, where $\mathrm{y}, \mathrm{t}, \mathbf{c} \subset \mathbf{x}$, if there exists two proxies $\mathbf{w}, \mathbf{n} \subset \mathbf{x}$ and $\mathbf{b} \subset \mathbf{x}$, none of them overlapping nor containing variables from the previous subsets, s.t.:*

i) $\mathbf{w}$ *is conditionally independent of* $(\mathrm{t}, \mathbf{n})$ *given $\mathbf{b}$, $\mathbf{z}$ and $\mathbf{c}$. That is, $\mathbf{w} \perp\!\!\!\perp (\mathrm{t}, \mathbf{n})|\, \mathbf{b}, \mathbf{z}, \mathbf{c}$.*

ii) $\mathbf{n}$ *is conditionally independent of* y *given t, $\mathbf{b}$, $\mathbf{z}$ and $\mathbf{c}$. That is, $\mathrm{y} \perp\!\!\!\perp \mathbf{n}|\, \mathrm{t}, \mathbf{b}, \mathbf{z}, \mathbf{c}$.*

iii) $(\mathbf{b}, \mathbf{z})$ *forms a valid adjustment set for the query $p(\mathrm{y}|\, do(\mathrm{t}), \mathbf{c})$. That is, given $\mathbf{c}$, they are independent of t after severing any incoming edges to it, $\mathrm{t} \perp\!\!\!\perp_{\mathcal{G}_{\overline{\mathrm{t}}}} (\mathbf{b}, \mathbf{z})|\, \mathbf{c}$, and they block every backdoor path from t to y.*

iv) $\mathbf{z}$ *is complete given $\mathbf{n}$ for almost all t, $\mathbf{b}$, and $\mathbf{c}$,*

v) $\tilde{\mathbf{z}}$ *is complete given $\mathbf{w}$ for almost all $\mathbf{b}$ and $\mathbf{c}$,*

*and the following regularity conditions also hold:*

vi) $\iint \tilde{p}(\tilde{\mathbf{z}}|\, \mathbf{w}, \mathbf{b}, \mathbf{c}) \tilde{p}(\mathbf{w}|\, \tilde{\mathbf{z}}, \mathbf{b}, \mathbf{c}) \, d\tilde{\mathbf{z}} \, d\mathbf{w} < \infty$ *for all $\mathbf{b}$, $\mathbf{c}$, and*

vii) $\int \tilde{p}(\mathrm{y}|\, \mathrm{t}, \mathbf{b}, \tilde{\mathbf{z}}, \mathbf{c})^2 \tilde{p}(\tilde{\mathbf{z}}|\, \mathbf{b}, \mathbf{c}) \, d\tilde{\mathbf{z}} < \infty$ *for all t, $\mathbf{b}$, and $\mathbf{c}$.*

*Proof.* First, note that the first three independence assumptions hold for both models, $\mathcal{M}$ and $\tilde{\mathcal{M}}$, as they induce the same causal graph. Following the same arguments as Miao et al. [41, Proposition 1], we have that assumptions **v)**, **vi)**, and **vii)** guarantee the existence of a function $\tilde{h}$ such that it solves the integral equation over $\tilde{\mathcal{M}}$,

$$\tilde{p}(\mathrm{y} \mid \mathrm{t}, \mathbf{b}, \tilde{\mathbf{z}}, \mathbf{c}) = \int \tilde{h}(\mathrm{y}, \mathrm{t}, \mathbf{b}, \mathbf{w}, \mathbf{c}) \tilde{p}(\mathbf{w} \mid \mathbf{b}, \tilde{\mathbf{z}}, \mathbf{c}) \, d\mathbf{w} \,, \tag{6}$$

since assumption **vi)** ensures that the conditional expectation operator is compact [8], assumption **v)** that all square-integrable functions are in the image of the operator (i.e., the operator is surjective), and assumption **vii)** that $\tilde{p}(\mathrm{y}|\, \mathrm{t}, \mathbf{b}, \tilde{\mathbf{z}}, \mathbf{c})$ is indeed part of the image.

We can show that $\tilde{h}$ also solves a similar integral equation, this time over the other SCM, $\mathcal{M}$, as follows:

$$\begin{aligned} p(\mathrm{y} \mid \mathrm{t}, \mathbf{b}, \mathbf{n}, \mathbf{c}) &= \tilde{p}(\mathrm{y} \mid \mathrm{t}, \mathbf{b}, \mathbf{n}, \mathbf{c}) && [\textit{equal marginals}] && (7) \\ &= \int \tilde{p}(\mathrm{y} \mid \mathrm{t}, \mathbf{b}, \mathbf{n}, \tilde{\mathbf{z}}, \mathbf{c}) \tilde{p}(\tilde{\mathbf{z}} \mid \mathrm{t}, \mathbf{b}, \mathbf{n}, \mathbf{c}) \, d\tilde{\mathbf{z}} && [\textit{augment with } \tilde{\mathbf{z}}] && (8) \\ &= \int \tilde{p}(\mathrm{y} \mid \mathrm{t}, \mathbf{b}, \tilde{\mathbf{z}}, \mathbf{c}) \tilde{p}(\tilde{\mathbf{z}} \mid \mathrm{t}, \mathbf{b}, \mathbf{n}, \mathbf{c}) \, d\tilde{\mathbf{z}} && [\textit{assumption } \textbf{\textit{ii)}}] && (9) \\ &= \iint \tilde{h}(\mathrm{y}, \mathrm{t}, \mathbf{b}, \mathbf{w}, \mathbf{c}) \tilde{p}(\mathbf{w} \mid \mathbf{b}, \tilde{\mathbf{z}}, \mathbf{c}) \tilde{p}(\tilde{\mathbf{z}} \mid \mathrm{t}, \mathbf{b}, \mathbf{n}, \mathbf{c}) \, d\tilde{\mathbf{z}} \, d\mathbf{w} && [\textit{plug Eq. 6}] && (10) \\ &= \iint \tilde{h}(\mathrm{y}, \mathrm{t}, \mathbf{b}, \mathbf{w}, \mathbf{c}) \tilde{p}(\mathbf{w} \mid \mathbf{b}, \tilde{\mathbf{z}}, \mathrm{t}, \mathbf{n}, \mathbf{c}) \tilde{p}(\tilde{\mathbf{z}} \mid \mathrm{t}, \mathbf{b}, \mathbf{n}, \mathbf{c}) \, d\tilde{\mathbf{z}} \, d\mathbf{w} && [\textit{assumption } \textbf{\textit{i)}}] && (11) \end{aligned}$$

$$= \int \tilde{h}(\mathrm{y}, \mathrm{t}, \mathbf{b}, \mathbf{w}, \mathbf{c}) p(\mathbf{w} \mid \mathrm{t}, \mathbf{b}, \mathbf{n}, \mathbf{c}) \, \mathrm{d}\mathbf{w} \,. \qquad \textit{[equal marginals]} \qquad (12)$$

Note that Eq. 12 is a Fredholm equation of the first kind that is implicitly solved by modeling the observational data. Similarly, we can relate the expression for the interventional distribution of both models:

$$\tilde{p}(\mathrm{y} \mid \mathrm{do}(\mathrm{t}), \mathbf{c}) = \int \tilde{p}(\mathrm{y} \mid \mathrm{do}(\mathrm{t}), \mathbf{b}, \tilde{\mathbf{z}}, \mathbf{c}) \tilde{p}(\mathbf{b}, \tilde{\mathbf{z}} \mid \mathbf{c}) \, \mathrm{d}\mathbf{b} \, \mathrm{d}\tilde{\mathbf{z}} \qquad \textit{[augment and ass. \textbf{iii})]} \qquad (13)$$

$$= \int \tilde{p}(\mathrm{y} \mid \mathrm{t}, \mathbf{b}, \tilde{\mathbf{z}}, \mathbf{c}) \tilde{p}(\mathbf{b}, \tilde{\mathbf{z}} \mid \mathbf{c}) \, \mathrm{d}\mathbf{b} \, \mathrm{d}\tilde{\mathbf{z}} \qquad \textit{[backdoor criterion]} \qquad (14)$$

$$= \iint \tilde{h}(\mathrm{y}, \mathrm{t}, \mathbf{b}, \mathbf{w}, \mathbf{c}) \tilde{p}(\mathbf{w} \mid \mathbf{b}, \tilde{\mathbf{z}}, \mathbf{c}) \tilde{p}(\mathbf{b}, \tilde{\mathbf{z}} \mid \mathbf{c}) \, \mathrm{d}\mathbf{b} \, \mathrm{d}\mathbf{w} \, \mathrm{d}\tilde{\mathbf{z}} \qquad \textit{[plug Eq. 6]} \qquad (15)$$

$$= \int \tilde{h}(\mathrm{y}, \mathrm{t}, \mathbf{b}, \mathbf{w}, \mathbf{c}) p(\mathbf{b}, \mathbf{w} \mid \mathbf{c}) \, \mathrm{d}\mathbf{b} \, \mathrm{d}\mathbf{w} \qquad \textit{[equal marginals]} \qquad (16)$$

$$= p(\mathrm{y} \mid \mathrm{do}(\mathrm{t}), \mathbf{c}) \,, \qquad (17)$$

where the last equality is a consequence of Eq. 12 as we will show now. More specifically, we have that

$$p(\mathrm{y} \mid \mathrm{t}, \mathbf{b}, \mathbf{n}, \mathbf{c}) = \int \tilde{h}(\mathrm{y}, \mathrm{t}, \mathbf{b}, \mathbf{w}, \mathbf{c}) p(\mathbf{w} \mid \mathrm{t}, \mathbf{b}, \mathbf{n}, \mathbf{c}) \, \mathrm{d}\mathbf{w} \qquad \textit{[Eq. 12]} \qquad (18)$$

$$= \iint \tilde{h}(\mathrm{y}, \mathrm{t}, \mathbf{b}, \mathbf{w}, \mathbf{c}) p(\mathbf{w} \mid \mathbf{b}, \mathbf{z}, \mathrm{t}, \mathbf{n}, \mathbf{c}) p(\mathbf{z} \mid \mathrm{t}, \mathbf{b}, \mathbf{n}, \mathbf{c}) \, \mathrm{d}\mathbf{w} \, \mathrm{d}\mathbf{z} \,, \qquad \textit{[augment with \textbf{z}]} \qquad (19)$$

$$= \iint \tilde{h}(\mathrm{y}, \mathrm{t}, \mathbf{b}, \mathbf{w}, \mathbf{c}) p(\mathbf{w} \mid \mathbf{b}, \mathbf{z}, \mathbf{c}) p(\mathbf{z} \mid \mathrm{t}, \mathbf{b}, \mathbf{n}, \mathbf{c}) \, \mathrm{d}\mathbf{w} \, \mathrm{d}\mathbf{z} \,. \qquad \textit{[assumption \textbf{i})]} \qquad (20)$$

Similarly, we have that

$$p(\mathrm{y} \mid \mathrm{t}, \mathbf{b}, \mathbf{n}, \mathbf{c}) = \int p(\mathrm{y} \mid \mathrm{t}, \mathbf{b}, \mathbf{n}, \mathbf{z}, \mathbf{c}) p(\mathbf{z} \mid \mathrm{t}, \mathbf{b}, \mathbf{n}, \mathbf{c}) \, \mathrm{d}\mathbf{z} \qquad \textit{[augment with \textbf{z}]} \qquad (21)$$

$$= \int p(\mathrm{y} \mid \mathrm{t}, \mathbf{b}, \mathbf{z}, \mathbf{c}) p(\mathbf{z} \mid \mathrm{t}, \mathbf{b}, \mathbf{n}, \mathbf{c}) \, \mathrm{d}\mathbf{z} \,. \qquad \textit{[assumption \textbf{ii})]} \qquad (22)$$

Now, equating both expressions we have that

$$0 = \iint \left\{ p(\mathrm{y} \mid \mathrm{t}, \mathbf{b}, \mathbf{z}, \mathbf{c}) - \int \tilde{h}(\mathrm{y}, \mathrm{t}, \mathbf{b}, \mathbf{w}, \mathbf{c}) p(\mathbf{w} \mid \mathbf{b}, \mathbf{z}, \mathbf{c}) \, \mathrm{d}\mathbf{w} \right\} p(\mathbf{z} \mid \mathrm{t}, \mathbf{b}, \mathbf{n}, \mathbf{c}) \, \mathrm{d}\mathbf{z} \,, \qquad (23)$$

which, due to assumption **iv)**, implies that

$$p(\mathrm{y} \mid \mathrm{t}, \mathbf{b}, \mathbf{z}, \mathbf{c}) \overset{\text{a.e.}}{=} \int \tilde{h}(\mathrm{y}, \mathrm{t}, \mathbf{b}, \mathbf{w}, \mathbf{c}) p(\mathbf{w} \mid \mathbf{b}, \mathbf{z}, \mathbf{c}) \, \mathrm{d}\mathbf{w} \,. \qquad (24)$$

Finally, putting all together we see that we can write the interventional distribution of the original model using $\tilde{h}$,

$$p(\mathrm{y} \mid \mathrm{do}(\mathrm{t}), \mathbf{c}) = \iint p(\mathrm{y} \mid \mathrm{do}(\mathrm{t}), \mathbf{b}, \mathbf{z}, \mathbf{c}) p(\mathbf{b}, \mathbf{z} \mid \mathbf{c}) \, \mathrm{d}\mathbf{b} \, \mathrm{d}\mathbf{z} \qquad \textit{[augment and assumption \textbf{iii})]} \qquad (25)$$

$$= \iint p(\mathrm{y} \mid \mathrm{t}, \mathbf{b}, \mathbf{z}, \mathbf{c}) p(\mathbf{b}, \mathbf{z} \mid \mathbf{c}) \, \mathrm{d}\mathbf{b} \, \mathrm{d}\mathbf{z} \qquad \textit{[backdoor criterion]} \qquad (26)$$

$$= \iint \tilde{h}(\mathrm{y}, \mathrm{t}, \mathbf{b}, \mathbf{w}, \mathbf{c}) p(\mathbf{w} \mid \mathbf{b}, \mathbf{z}, \mathbf{c}) p(\mathbf{b}, \mathbf{z} \mid \mathbf{c}) \, \mathrm{d}\mathbf{b} \, \mathrm{d}\mathbf{z} \, \mathrm{d}\mathbf{w} \qquad \textit{[Eq. 24]} \qquad (27)$$

$$= \int \tilde{h}(\mathrm{y}, \mathrm{t}, \mathbf{b}, \mathbf{w}, \mathbf{c}) p(\mathbf{b}, \mathbf{w} \mid \mathbf{c}) \, \mathrm{d}\mathbf{b} \, \mathrm{d}\mathbf{w} \,, \qquad \textit{[equal marginals]} \qquad (28)$$

which justifies the last equality in Eq. 17. □

Using a causal graph similar to the one presented by Miao et al. [41], we now provide some intuition on the semantics of each random variable in Prop. A.2. More specifically, consider the causal graph that we depict in Fig. 5, and say that we

want to check if the causal query $p(\mathbf{y}|\,\mathrm{do}(\mathbf{t}))$ is identifiable (note that this the same query as in Prop. A.2 but with $\mathbf{c} = \emptyset$). As it is common in the causal inference literature [49, 60], t and y represent the treatment and outcome random variables. More specific to Prop. A.2 are $\mathbf{w}$ and $\mathbf{n}$. Here, $\mathbf{w}$ is a proxy variable whose role is that of distinguishing the information from $\mathbf{z}$ and other variables, to reconstruct the information of $\mathbf{z}$ and block the backdoor path that $\mathbf{z}$ would usually block. Similarly, the variable $\mathbf{n}$ is another proxy variable which, in this case, serves the purpose of verifying that the substitute formed with $\mathbf{w}$ is indeed a good one. Finally, the variable $\mathbf{b}$ serves the purpose of blocking all the remaining backdoor paths that $\mathbf{z}$ may not block, so that we can apply the backdoor criterion.

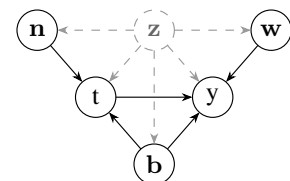

Moreover, note that for all interventional queries we let $\mathbf{c}$ be the empty set, similar to the results proved by Miao et al. [41] and Wang and Blei [66]. We will consider cases when $\mathbf{c}$ is not empty later in §A.3 to prove counterfactual identifiability. Note also that Prop. A.2 reduces to previous results when $\mathbf{c} = \mathbf{b} = \emptyset$.

Figure 5: Example for which Prop. A.2 applies, and where $\mathbf{b} \neq \emptyset$.

We now turn our attention towards proving Cor. 4.2, i.e., towards broadening the concept of query identifiability by introducing Prop. A.2 as a base case of do-calculus. To this end, we introduce the concept of a *hedge* which will be use later, but we still strongly recommend reading the work by Shpitser and Pearl [59].

**Definition 5** (Hedge, [59, Def. 6]). *Let* $\mathbf{y}, \mathbf{t} \subset \mathbf{x}$ *be disjoint sets of variables in* $\mathcal{G}$. *Let* $F$, $F'$ *be* $\mathbf{r}$-*rooted C-forests (see [59, Def. 5]) such that* $F \cap \mathbf{t} \neq \emptyset$, $F' \cap \mathbf{t} = \emptyset$, $F' \subset F$, *and* $\mathbf{r}$ *is a subset of the ancestors of* $\mathbf{y}$ *after severing the incoming edges of* $\mathbf{t}$. *Then* $F$ *and* $F'$ *form a hedge for* $p(\mathbf{y}|\,\mathrm{do}(\mathbf{t}))$ *in* $\mathcal{G}$.

**Corollary 4.2.** *An interventional query is identifiable if, using do-calculus, it can be reduced to a combination of observational queries and identifiable interventional queries in the sense of Prop. 4.1.*

*Proof.* With the additional notion of proxy-identifiability provided by Prop. A.2 (informally presented in Prop. 4.1), the result is just a consequence of applying the identifiability algorithm provided by Shpitser and Pearl [59]. See also [23, 62] for other references.

Since the do-calculus rules are complete in the classical sense of identifibiability, a query is not identifiable if the aforementioned algorithm yields a `FAIL` status (i.e., it executes line 5 of Figure 3 in [59]). If that is the case, then it means that, at the specific recursive call for which the algorithm failed, the local graph $\mathcal{G}$ contains a hedge and the interventional query $p(\mathbf{y}|\,\mathrm{do}(\mathbf{t}))$ is not identifiable in the classical sense.

Crucially, this hedge $(F, F')$ expresses the inability of identifying an interventional query of the form $p(\mathbf{r}|\,\mathrm{do}(\mathbf{t}'))$ where the root $\mathbf{r}$ is a subset of ancestors of $\mathbf{y}' \subseteq \mathbf{y}$ and $\mathbf{t}' \subseteq \mathbf{t}$. Then, this local query can still be proxy-identifiable if Prop. A.2 can be applied, and thus we can continue running the identification algorithm.

The stated result is then a consequence of successfully applying the logic above each time we find a `FAIL` status, yielding a final `FAIL` status otherwise. $\square$

To be even more explicit regarding the identifiability of the queries proven in corollary above, let us call $\mathcal{M}$ the original SCM as usual, and $\tilde{\mathcal{M}}$ another SCM inducing the same causal graph as $\mathcal{M}$ and which matches the observational marginal distribution of $\mathcal{M}$, i.e., $p(\mathbf{x}) \overset{\text{a.e.}}{=} \tilde{p}(\mathbf{x})$. Then, the output of the identifiability algorithm from the corollary above *for both SCMs* will be two identical expressions `EXP` composed of sum, integrals, and products of observational quantities (i.e., marginals and conditionals of subsets of $\mathbf{x}$) as well as proxy-identifiable queries of the form $p(\mathbf{y}|\,\mathrm{do}(\mathbf{t}))$ as in Prop. A.2. Therefore,

$$Q(\mathcal{M}) = \mathrm{EXP}(\mathcal{M}) = \mathrm{EXP}(\tilde{\mathcal{M}}) = Q(\tilde{\mathcal{M}}), \tag{29}$$

where the second equality is a consequence of both SCMs having equal observational distributions (and thus any other quantity than can derived exclusively from $p(\mathbf{x})$) and of applying Prop. A.2 for any interventional query that appears in the expression.

**Illustrative example.** To understand the implications of Prop. 4.1 and Cor. 4.2, consider the causal graph in Fig. 6, and suppose we want to compute $Q(\mathcal{M}) = p(\mathbf{y}_1|\,\mathrm{do}(\mathbf{t}))$. Then, we can proceed as usual and apply the rules of probability theory and do-calculus to rewrite $Q(\mathcal{M})$ as

$$Q(\mathcal{M}) = \int p(\mathbf{y}_1 \mid \mathbf{t}, \mathbf{y}_2) p(\mathbf{y}_2 \mid \mathrm{do}(\mathbf{t}))\,\mathrm{d}\mathbf{y}_2. \tag{30}$$

As a result, the identifiability of $p(\mathbf{y}_2|\,\mathrm{do}(\mathbf{t}))$ implies that of $Q(\mathcal{M})$. We can then devise a few different scenarios:

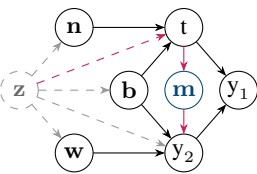

Figure 6: Causal graph for which the presence or absence of some parts render $p(y_1 | do(t))$ identifiable using do-calculus. Else, Prop. 4.1 yields identifiability if **w** and **n** are informative proxies.

1. If there is no edge from **z** to t, i.e., $t \notin ch(\mathbf{z})$, then the backdoor criterion [49] holds for $\{\mathbf{n}, \mathbf{b}\} = pa(t) \subset \mathbf{x}$ and both $p(y_1 | do(t))$ and $p(y_2 | do(t))$ are identifiable.

2. If there exists a mediator **m** between t and $y_2$, we can apply the front-door adjustment [49] and both $p(y_1 | do(t))$ and $p(y_2 | do(t))$ are identifiable.

3. If $y_2$ is not caused by t, then we have that $p(y_2 | do(t)) = p(y_2)$ and both queries are identifiable.

4. Otherwise, we can still render $p(y_2 | do(t))$ identifiable if **w** and **n** yield sufficient information about **z** (intuitively, if the posterior of **z** changes enough as we change **w** and **n**; see Def. 4) and we can hence apply Prop. 4.1.

The example above nicely illustrates how Prop. 4.1 complements do-calculus: if we find a query unidentifiable due to reaching a dead end with do-calculus—in this case, $p(y_2 | do(t))$—then Prop. 4.1 provides an additional case for which the query can still be made identifiable. Moreover, this case clearly shows how Prop. 4.1 extends prior results as these *did not allow* for common observable ancestors between outcome and treatment [41, 66]. Nevertheless, note that Prop. 4.1 provides only sufficient conditions for identifiability, and there could exist identifiable queries which do not comply with the requirements of the proposition.

**Corollary 4.3.** *If DeCaFlow induces the same causal graph as $\mathcal{M}$ and $p_{\mathcal{M}}(\mathbf{x}) \stackrel{a.e.}{=} p_{\boldsymbol{\theta}}(\mathbf{x})$, then it correctly estimates any query identifiable in the sense of Cor. 4.2.*

*Proof.* The proof is a direct consequence of the corollary above and the fact that we can interpret DeCaFlow as a dense parametric family of confounded SCMs inducing the same causal graph as $\mathcal{M}$ (similar to the interpretation of Javaloy et al. [25] as bijective SCMs) by considering the triplet $\mathcal{M}_{\boldsymbol{\theta}} := (T_{\boldsymbol{\theta}}^{-1}, P_{\mathbf{u}}, P_{\mathbf{z}})$, where $T_{\boldsymbol{\theta}}^{-1}$ is the inverse of the generative network that transforms **u** into **x** given **z**. This family being dense is a consequence of the generative networks forming a family of universal density approximators [25, 44]. $\square$

To be completely exhaustive, in the following we explore the general proposition Prop. A.2 on all scenarios where t and y may or may not be directly caused by the hidden confounder, as we show in the following subsections.

### A.2.1 Fully hidden-confounded case

In the case where both variables are children of **z**, we must see whether we can apply do-calculus with Prop. A.2 as an additional base case, as described in Cor. 4.2.

### A.2.2 Hidden-unconfounded case

Assume the case where neither t nor y are children of the hidden confounder, i.e., $y, t \notin ch(\mathbf{z})$. In this case, the proof of Prop. A.2 can be simplified and drop the requirement of finding valid proxy variables.

**Corollary A.3.** *Given two SCMs $\mathcal{M} := (\mathbf{f}, P_{\mathbf{u}}, P_{\mathbf{z}})$ and $\tilde{\mathcal{M}} := (\tilde{\mathbf{f}}, P_{\tilde{\mathbf{u}}}, P_{\tilde{\mathbf{z}}})$, assume that they are Markov-equivalent—i.e., they induce the same causal graph—and coincide in their marginal distributions, $p(\mathbf{x}) \stackrel{a.e.}{=} \tilde{p}(\mathbf{x})$. If $y, t \notin ch(\mathbf{z})$, then, $p(\mathbf{y} | do(t), \mathbf{c}) = \tilde{p}(\mathbf{y} | do(t), \mathbf{c})$, where $y, t, \mathbf{c} \subset \mathbf{x}$.*

*Proof.* The proof follows directly by applying Prop. A.2 with the minimal subset $\mathbf{b} \subset pa(t) \setminus \{\mathbf{c}\}$ that blocks all the backdoor paths, and by noticing that in this case there is no need to use the variables **z** and $\tilde{\mathbf{z}}$. That is, we can go from Eq. 13 to Eq. 17 directly by using only **b** and the equal-marginals assumption:

$$\tilde{p}(\mathbf{y} | do(t), \mathbf{c}) = \int \tilde{p}(\mathbf{y} | do(t), \mathbf{b}, \mathbf{c})\tilde{p}(\mathbf{b} | \mathbf{c}) \, d\mathbf{b} \tag{31}$$

$$= \int \tilde{p}(\mathbf{y} | t, \mathbf{b}, \mathbf{c})\tilde{p}(\mathbf{b} | \mathbf{c}) \, d\mathbf{b} \tag{32}$$

$$= \int p(y \mid t, \mathbf{b}, \mathbf{c}) p(\mathbf{b} \mid \mathbf{c}) \, d\mathbf{b} \tag{33}$$

$$= p(y \mid \mathrm{do}(t), \mathbf{c}) \,. \tag{34}$$

$$\square$$

Even though we can leverage and simplify Prop. A.2 as shown above, it is worth remarking that, for this particular case, the model identifiability results described in §A.1 are stronger, as it provides results on the identifiability of the causal generators and exogenous distributions, and therefore of any causal query derived from them.

### A.2.3 Confounded outcome case

For the case where only the outcome variable is a child of the hidden confounder, we can apply a similar reasoning as we did in the previous case, although this time we cannot leverage the stronger results from Javaloy et al. [25]. More specifically:

**Corollary A.4.** *Given two SCMs $\mathcal{M} := (\mathbf{f}, P_\mathbf{u}, P_\mathbf{z})$ and $\tilde{\mathcal{M}} := (\tilde{\mathbf{f}}, P_{\tilde{\mathbf{u}}}, P_{\tilde{\mathbf{z}}})$, assume that they are Markov-equivalent—i.e., they induce the same causal graph—and coincide in their marginal distributions, $p(\mathbf{x}) \overset{a.e.}{=} \tilde{p}(\mathbf{x})$. Assume that $t \notin \mathrm{ch}(\mathbf{z})$. Then, $p(y \mid \mathrm{do}(t), \mathbf{c}) = \tilde{p}(y \mid \mathrm{do}(t), \mathbf{c})$, where $y, t, \mathbf{c} \subset \mathbf{x}$.*

*Proof.* The proof is identical to that of Cor. A.3. $\square$

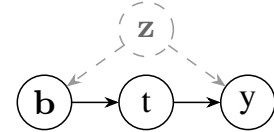

Figure 7: Example of a front-door causal.

**Front-door example.** While the proof above is trivial given the previous results, it is worth stressing the importance of modeling the hidden confounder as we do in this work with DeCaFlow. As an example, consider the SCM depicted in Fig. 7, where we have that the outcome is directly confounded by $\mathbf{z}$, while t is not. In this case, DeCaFlow can correctly estimate the causal effects of $\mathbf{b}$ and t on y, i.e., to correctly estimate $p(y \mid \mathrm{do}(t))$ and $p(y \mid \mathrm{do}(\mathbf{b}))$, using $\tilde{\mathbf{z}}$ to model the influence of $\mathbf{b}$ onto y that is not explained through t. Other models that do not model $\mathbf{z}$—e.g., an unaware CNF [25]—would be able to match the observed marginal distribution (as they are universal density approximators) and therefore to estimate $p(y \mid \mathrm{do}(\mathbf{b}))$ (as it is identifiable through the mediator t using the front-door criterion), yet they would necessarily fail to estimate $p(y \mid \mathrm{do}(t))$, since they assume that $y \perp\!\!\!\perp \mathbf{b} \mid t$ yet we know that $y \not\perp\!\!\!\perp \mathbf{b} \mid t$ in the true model. In other words, an unaware CNF would hold that $p(y \mid \mathrm{do}(t)) = p(y \mid t)$ which is clearly false by looking at Fig. 7.

To be even more explicit, in this case we would have a data-generating process that factorizes as

$$\tilde{p}(\mathbf{b}, t, y, \tilde{\mathbf{z}}) = \tilde{p}(\tilde{\mathbf{z}}) \tilde{p}(\mathbf{b} \mid \tilde{\mathbf{z}}) \tilde{p}(t \mid \mathbf{b}) \tilde{p}(y \mid t, \tilde{\mathbf{z}}) \,, \tag{35}$$

and hence the estimated interventional distribution from DeCaFlow matches the true one:

$$p(y \mid \mathrm{do}(t)) = \int p(y \mid t, \mathbf{b}) p(\mathbf{b}) \, d\mathbf{b} \qquad [\textit{\textbf{b} forms a valid adjustment set}] \tag{36}$$

$$= \int \left\{ \int \tilde{p}(y \mid t, \mathbf{b}, \tilde{\mathbf{z}}) \tilde{p}(\tilde{\mathbf{z}} \mid t, \mathbf{b}) \, d\tilde{\mathbf{z}} \right\} \tilde{p}(\mathbf{b}) \, d\mathbf{b} \qquad [\textit{Factorization and eq. marginals}] \tag{37}$$

$$= \iint \tilde{p}(y \mid t, \tilde{\mathbf{z}}) \tilde{p}(\tilde{\mathbf{z}} \mid \mathbf{b}) \tilde{p}(\mathbf{b}) \, d\mathbf{b} \, d\tilde{\mathbf{z}} \qquad [\textit{Factorization in Eq. 35}] \tag{38}$$

$$= \int \tilde{p}(y \mid t, \tilde{\mathbf{z}}) \tilde{p}(\tilde{\mathbf{z}}) \, d\tilde{\mathbf{z}} \qquad [\textit{marginalize \textbf{b}}] \tag{39}$$

$$= \tilde{p}(y \mid \mathrm{do}(t)) \,. \tag{40}$$

### A.2.4 Hidden-confounded treatment case

When only the treatment variable t is a child of $\mathbf{z}$, we can face two different scenarios: **i)** we find a valid adjustment set $\mathbf{b}$ blocking all backdoor paths, in which case we can reason just as in the other partially hidden-confounded case, and **ii)** we cannot, and then rely on do-calculus and the identifiability w.r.t. $\mathbf{b}$. For example, if $\mathbf{b}$ happens to be a parent of y which is directly caused by the treatment variable t and the hidden confounder $\mathbf{z}$ as in Fig. 8, we cannot find a valid adjustment set for the causal query, but it may still serve us if we can identify the same query with the adjustment set as outcome variable.

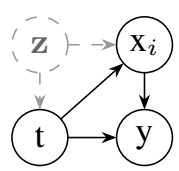

Figure 8: Case with no valid adjustment set.

**Corollary A.5.** *Given two SCMs* $\mathcal{M} := (\mathbf{f}, P_{\mathbf{u}}, P_{\mathbf{z}})$ *and* $\tilde{\mathcal{M}} := (\tilde{\mathbf{f}}, P_{\tilde{\mathbf{u}}}, P_{\tilde{\mathbf{z}}})$, *assume that they are Markov-equivalent—i.e., they induce the same causal graph—and coincide in their marginal distributions,* $p(\mathbf{x}) \overset{a.e.}{=} \tilde{p}(\mathbf{x})$. *If* $\mathbf{y} \notin \mathrm{ch}(\mathbf{z})$ *then,* $p(\mathbf{y} | do(\mathbf{t}), \mathbf{c}) = \tilde{p}(\mathbf{y} | do(\mathbf{t}), \mathbf{c})$, *where* $\mathbf{y}, \mathbf{t}, \mathbf{c} \subset \mathbf{x}$ *if there exists* $\mathbf{b} \subset \mathbf{x}$ *not containing variables from the previous subsets, such that one of the following two conditions are true:*

i) $\mathbf{b}$ *forms a valid adjustment set for the query* $p(\mathbf{y} | do(\mathbf{t}), \mathbf{c})$.

ii) $\mathbf{b}$ *blocks all backdoor paths and the query* $p(\mathbf{b} | do(\mathbf{t}), \mathbf{c})$ *is identifiable.*

*Proof.* If condition **i)** holds, then we have a valid adjustment set, and the proof is identical to that of Cor. A.3.

Otherwise, if condition **ii)** holds, we have that the interventional query on y equals the observational query when conditioned on $\mathbf{b}$, but that now $\mathbf{b}$ is not independent of $do(\mathbf{t})$, i.e.,

$$\tilde{p}(\mathbf{y} | do(\mathbf{t}), \mathbf{c}) = \int \tilde{p}(\mathbf{y} | do(\mathbf{t}), \mathbf{b}, \mathbf{c})\tilde{p}(\mathbf{b} | do(\mathbf{t}), \mathbf{c})\, d\mathbf{b} \tag{41}$$

$$= \int \tilde{p}(\mathbf{y} | \mathbf{t}, \mathbf{b}, \mathbf{c})\tilde{p}(\mathbf{b} | do(\mathbf{t}), \mathbf{c})\, d\mathbf{b} \tag{42}$$

$$= \int p(\mathbf{y} | \mathbf{t}, \mathbf{b}, \mathbf{c})p(\mathbf{b} | do(\mathbf{t}), \mathbf{c})\, d\mathbf{b} \tag{43}$$

$$= p(\mathbf{y} | do(\mathbf{t}), \mathbf{c})\,, \tag{44}$$

where we needed to use that the query $p(\mathbf{b} | do(\mathbf{t}), \mathbf{c})$ is identifiable in the third equality. $\square$

### A.2.5 Napkin example

Finally, we want to show one last illustrative example where DeCaFlow provides correct estimates of a causal query that is identifiable by the do-calculus, but neither the backdoor nor the front-door criteria are applicable. While redundant (as the query is identifiable in the classical sense, and then Cor. 4.2 applies), we believe it can be a good exercise to convince the reader. Namely, the graph of Fig. 9 appears as the napkin graph in Pearl and Mackenzie [45, Fig. 7.5]. What is particularly interesting in this graph is that w is not a valid adjustment set since, despite blocking the backdoor path from t to y through $\mathbf{b}$, it forms a collider of $\mathbf{z}_1$ and $\mathbf{z}_2$.

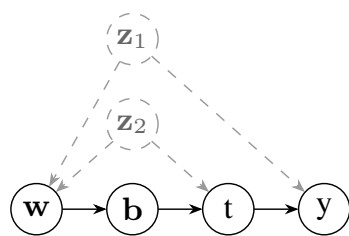

Figure 9: Napkin causal graph [45].

However, $\mathbf{z}_1$ only affects the outcome and $\mathbf{z}_2$ only affects the treatment. Following from our previous results, the causal effect from t to y should be correctly estimated by DeCaFlow. Here, we show that this is the case. First, let us express the causal query of interest in another form applying do-calculus:

$$p(\mathbf{y} | do(\mathbf{t})) = p(\mathbf{y} | do(\mathbf{y}), do(\mathbf{t})) = \quad [\textit{Rule 3 of do-calculus since } \mathbf{y} \perp\!\!\!\perp_{\mathcal{G}_{\bar{\mathbf{t}},\bar{\mathbf{b}}}} \mathbf{b} | \mathbf{t}] \tag{45}$$

$$= p(\mathbf{y} | \mathbf{t}, do(\mathbf{b})) = \quad [\textit{Rule 2 of do-calculus } \mathbf{y} \perp\!\!\!\perp_{\mathcal{G}_{\bar{\mathbf{b}},\underline{\mathbf{t}}}} \mathbf{t} | \mathbf{b}] \tag{46}$$

$$= \frac{p(\mathbf{y}, \mathbf{t} | do(\mathbf{b}))}{p(\mathbf{t} | do(\mathbf{b}))} \quad [\textit{Conditional probability}] \tag{47}$$

Once we have this expression, let us work on the numerator, considering that DeCaFlow is Markov-equivalent with the graph in Fig. 9:

$$p(\mathbf{y}, \mathbf{t} | do(\mathbf{b})) = \int p(\mathbf{y}, \mathbf{t} | \mathbf{b}, \mathbf{w})p(\mathbf{w})\, d\mathbf{w} \quad [\textit{Backdoor criterion}] \tag{48}$$

$$= \iiint \tilde{p}(\mathbf{y}, \mathbf{t}, \tilde{\mathbf{z}}_1, \tilde{\mathbf{z}}_2 | \mathbf{b}, \mathbf{w})p(\mathbf{w})\, d\mathbf{w}\, d\tilde{\mathbf{z}}_1\, d\tilde{\mathbf{z}}_2 \quad [\textit{Eq. marginals}] \tag{49}$$

$$= \iiint \tilde{p}(\mathbf{y}|\mathbf{t}, \tilde{\mathbf{z}}_1, \tilde{\mathbf{z}}_2, \mathbf{b}, \mathbf{w})\tilde{p}(\mathbf{t}|\tilde{\mathbf{z}}_1, \tilde{\mathbf{z}}_2, \mathbf{b}, \mathbf{w})\tilde{p}(\tilde{\mathbf{z}}_1, \tilde{\mathbf{z}}_2|\mathbf{w})p(\mathbf{w})\, d\mathbf{w}\, d\tilde{\mathbf{z}}_1\, d\tilde{\mathbf{z}}_2 \quad [\textit{Factorization}] \tag{50}$$

$$= \iiint \tilde{p}(\mathbf{y} | \mathbf{t}, \tilde{\mathbf{z}}_2)\tilde{p}(\mathbf{t} | \tilde{\mathbf{z}}_2, \mathbf{b})\tilde{p}(\tilde{\mathbf{z}}_1, \tilde{\mathbf{z}}_2 | \mathbf{w})p(\mathbf{w})\, d\mathbf{w}\, d\tilde{\mathbf{z}}_1\, d\tilde{\mathbf{z}}_2 \quad [\textit{Do-calculus rule 1}] \tag{51}$$

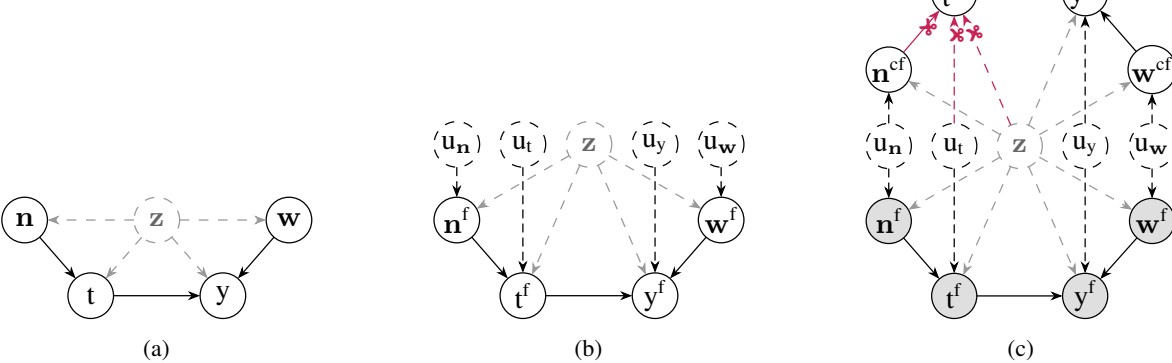

Figure 10: Example of the transition from **(a)** the regular depiction of a (confounded) SCM, to **(b)** an explicit SCM where the exogenous variables are drawn, and **(c)** a counterfactual twin SCM where the data-generating process is replicated in the "factual and counterfactual worlds". Figure **(c)** also depicts which nodes are observed and which are severed in order to compute a counterfactual query of the type $p(\mathrm{y}^{\mathrm{cf}}|\,\mathrm{do}(\mathrm{t}^{\mathrm{cf}}),\mathbf{x}^{\mathrm{f}})$.

$$= \int\int \tilde{p}(\mathrm{y}\mid \mathrm{t}, \tilde{\mathbf{z}}_2)\tilde{p}(\mathrm{t}\mid\tilde{\mathbf{z}}_2,\mathbf{b})\tilde{p}(\tilde{\mathbf{z}}_1,\tilde{\mathbf{z}}_2)\,\mathrm{d}\tilde{\mathbf{z}}_1\,\mathrm{d}\tilde{\mathbf{z}}_2 \qquad [\textit{Marginalize } \mathbf{w}] \qquad (52)$$

$$= \int\int \tilde{p}(\mathrm{y}\mid \mathrm{t}, \tilde{\mathbf{z}}_2)\tilde{p}(\mathrm{t}\mid\tilde{\mathbf{z}}_2,\mathbf{b})\tilde{p}(\tilde{\mathbf{z}}_1)\tilde{p}(\tilde{\mathbf{z}}_2)\,\mathrm{d}\tilde{\mathbf{z}}_1\,\mathrm{d}\tilde{\mathbf{z}}_2 \qquad [\tilde{\mathbf{z}}_1 \perp\!\!\!\perp_{\mathcal{G}} \mathbf{z}_2] \qquad (53)$$

$$= \int \tilde{p}(\mathrm{y}\mid \mathrm{t}, \tilde{\mathbf{z}}_2)\tilde{p}(\tilde{\mathbf{z}}_1)\,\mathrm{d}\tilde{\mathbf{z}}_1 \int \tilde{p}(\tilde{\mathbf{z}}_2)\tilde{p}(\mathrm{t}\mid\tilde{\mathbf{z}}_2,\mathbf{b})\,\mathrm{d}\tilde{\mathbf{z}}_2 \qquad [\textit{Separate integrals}] \qquad (54)$$

$$= \tilde{p}(\mathrm{y}\mid\mathrm{do}(\mathrm{t}))\,\tilde{p}(\mathrm{t}\mid\mathrm{do}(\mathbf{b})) \qquad [\textit{DeCaFlow estimate}] \qquad (55)$$

Note also that, as shown in Eq. 40, DeCaFlow correctly estimates $p(\mathrm{t}|\,\mathrm{do}(\mathbf{b}))$. Therefore, if we substitute Eq. 55 in Eq. 47, we have that

$$p(\mathrm{y}\mid\mathrm{do}(\mathrm{t})) = \frac{\tilde{p}(\mathrm{y}|\,\mathrm{do}(\mathrm{t}))\,p(\mathrm{t}|\,\mathrm{do}(\mathbf{b}))}{p(\mathrm{t}|\,\mathrm{do}(\mathbf{b}))} = \tilde{p}(\mathrm{y}\mid\mathrm{do}(\mathrm{t}))\,. \qquad (56)$$

That is, we have explicitly shown that DeCaFlow correctly estimates the true causal query $p(\mathrm{y}|\,\mathrm{do}(\mathrm{t}))$.

### A.3 COUNTERFACTUAL QUERY IDENTIFIABILITY

In this section, we show that counterfactual query identifiability is a direct result of the interventional query identifiability from the previous section.

In order to formally define counterfactuals, in this section we introduce the concept of counterfactual SCMs in a rather untraditional fashion. Namely, we combine the concepts of twin networks from Pearl [47] (which replicates the data-generating process) and that of counterfactual SCMs from Peters et al. [49] (which defines a counterfactual *prior* to the intervention) as follows:

**Definition 6** (Counterfactual twin SCM). Given a SCM $\mathcal{M} = (\mathbf{f}, P_{\mathbf{u}}, P_{\mathbf{z}})$, we define its counterfactual twin SCM as a SCM $\mathcal{M}^{\mathrm{cf}}$ where all structural equations are duplicated, and the exogenous noise is shared across replications, and where additionally one of the halves is observed ("the factual world"), and the other half is unobserved ("the counterfactual world").

We provide in Fig. 10 a more intuitive depiction on the construction of these counterfactual twin networks. From this definition, one can recover the counterfactual SCM defined by Peters et al. [49] by just focusing on the replicated part of the counterfactual twin network, and conditioning the exogenous noise and hidden confounder on the observed half, i.e., $(\mathbf{f}, P_{\mathbf{u}|\,\mathbf{x}^{\mathrm{f}}}, P_{\mathbf{z}|\,\mathbf{x}^{\mathrm{f}}})$. Similarly, one can compute the usual counterfactual query by performing an intervention on the counterfactual twin network, i.e., by replacing the intervened equations by the constant intervened value, and computing the query conditioned on the factual variables, $p(\mathrm{y}^{\mathrm{cf}}|\,\mathrm{do}(\mathrm{t}^{\mathrm{cf}}),\mathbf{x}^{\mathrm{f}})$. This is visually represented in Fig. 10c.

In order to prove query identifiability in the counterfactual setting, we need to use the following technical result regarding the completeness of a random variable:

**Lemma A.6.** *If a random variable* $\mathbf{z}$ *is complete given* $\mathbf{n}$ *for almost all* $\mathbf{b}$*, as given by* Def. 4*, then it is complete given* $\mathbf{n}$ *for almost all* $\mathbf{b}$ *and* $\mathbf{c}$*, where* $\mathbf{c}$ *is another continuous random variable.*

*Proof.* We prove this result by contradiction. Assume that the result does not hold, then there must exist a non-zero measure subset of the space of $\mathbf{b} \times \mathbf{c}$ for which there exists a square-integrable function $g(\cdot)$ such that $\int g(\mathbf{z}, \mathbf{b}, \mathbf{c}) p(\mathbf{z} | \mathbf{b}, \mathbf{c}, \mathbf{n}) \, \mathrm{d}\mathbf{z} = 0$ for almost all $\mathbf{n}$, but $g(\mathbf{z}, \mathbf{b}, \mathbf{c}) \neq 0$ for almost all $\mathbf{z}$.

Since this subset has positive measure, there must contain an $\varepsilon$-ball within. If we now focus on the $\mathbf{b}$-projection of this ball where we fix $\mathbf{c}$ to its value on the center, we have that it is a subset of non-zero measure in the space of $\mathbf{b}$ (as otherwise it would be zero-measure in the Cartesian-product measure), where the function $g(\cdot, \mathbf{c})$ breaks our initial assumption of the completeness of $\mathbf{z}$. Thus, we reach a contradiction. □

Given Def. 6, it is rather intuitive that, if a causal query is identifiable in a SCM $\mathcal{M}$, then it has to be identifiable in both halves of its induced counterfactual twin SCM $\mathcal{M}^{\mathrm{cf}}$, as they are identical. More importantly, we can now leverage again Prop. A.2, this time with $\mathbf{c} = \mathbf{x}^{\mathrm{f}}$, to prove counterfactual query identifiability whenever we have interventional query identifiability.

**Proposition A.7** (Counterfactual identifiability). *Given two SCMs* $\mathcal{M} := (\mathbf{f}, P_{\mathbf{u}}, P_{\mathbf{z}})$ *and* $\tilde{\mathcal{M}} := (\tilde{\mathbf{f}}, P_{\tilde{\mathbf{u}}}, P_{\tilde{\mathbf{z}}})$*, assume that they are Markov-equivalent—i.e., they induce the same causal graph—and that they coincide in their marginal distributions,* $p(\mathbf{x}) \overset{a.e.}{=} \tilde{p}(\mathbf{x})$*. Then, if a query* $p(\mathbf{y} | do(\mathbf{t}))$ *is identifiable in the sense of* Prop. A.2*, where* $\mathbf{y}, \mathbf{t} \subset \mathbf{x}$*, the query* $p(\mathbf{y}^{\mathrm{cf}} | do(\mathbf{t}^{\mathrm{cf}}), \mathbf{x}^{\mathrm{f}})$ *is also identifiable in the induced counterfactual twin SCM as long as the regularity conditions still hold, i.e., if:*

*i)* $\iint \tilde{p}(\tilde{\mathbf{z}} | \mathbf{w}, \mathbf{b}, \mathbf{c}) \tilde{p}(\mathbf{w} | \tilde{\mathbf{z}}, \mathbf{b}, \mathbf{c}) \, \mathrm{d}\tilde{\mathbf{z}} \, \mathrm{d}\mathbf{w} < \infty$ *for almost all* $\mathbf{b}$*,* $\mathbf{c}$*, and*

*ii)* $\int \tilde{p}(\mathbf{y} | \mathbf{t}, \mathbf{b}, \tilde{\mathbf{z}}, \mathbf{c})^2 \tilde{p}(\tilde{\mathbf{z}} | \mathbf{b}, \mathbf{c}) \, \mathrm{d}\tilde{\mathbf{z}} < \infty$ *for almost all* $\mathbf{t}$*,* $\mathbf{b}$*, and* $\mathbf{c}$*.*

*Proof.* We essentially need to prove that the independence and completeness assumptions keep holding when we add the factual covariate, $\mathbf{c} = \mathbf{x}^{\mathrm{f}}$.

For the independence, we need to show that, if we have a set of variables that fulfill the independence conditions from Prop. A.2, then this set of variables keeps holding them if we include $\mathbf{c} = \mathbf{x}^{\mathrm{f}}$. This is, however, easy to show since factual and counterfactual variables only have "tail-to-tail" dependencies, i.e., they are connected only through the shared exogenous variables. As a result, if two variables from the same half are conditionally independent given a third set of variables, conditioning on the other half cannot change this independence.

For the completeness, we need to show that introducing the factual variable retains the completeness assumed in Prop. A.2, which is direct to show using Lemma A.6. Specifically, it holds that

i) $\mathbf{z}$ is complete given $\mathbf{n}$ for almost all $\mathbf{t}$, $\mathbf{b}$, and $\mathbf{c}$, and

ii) $\tilde{\mathbf{z}}$ is complete given $\mathbf{w}$ for almost all $\mathbf{b}$ and $\mathbf{c}$.

Therefore, the requirements of Prop. A.2 hold when we append a factual variable to the twin network, and thus we can reapply all the results from the previous sections to the counterfactual cases. □

Once proven the result above, proving Cor. 4.3 is direct by following the exact same steps as we did in §A.2 to the counterfactual twin network instead of the original network.

It is important to note that, while the results above provide counterfactual identifiability whenever we have interventional identifiability, we still rely on how much of a good approximation the encoder is to the inverse of the decoder in the proposed DeCaFlow model. That is, the quality of the encoder determines how well we can perform the abduction step to compute counterfactuals. This consideration is unique to counterfactuals, as we just have to sample from the prior of $\mathbf{z}$ in the case of interventional queries.

# B   EXPERIMENTAL DETAILS AND ADDITIONAL RESULTS

## B.1   ABLATION STUDY

We conduct a simple ablation to understand the extent for which misspecifying the size of $\mathbf{z}$ affects DeCaFlow, as well as its sensitivity to the number of available proxies.

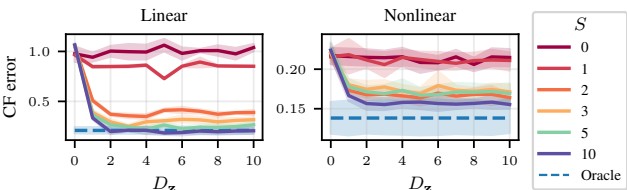
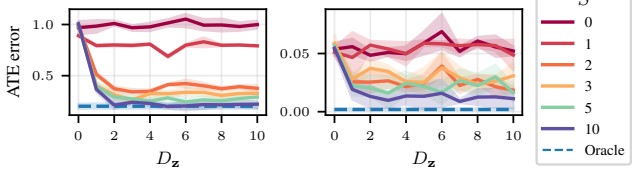

Figure 12: **Ablation study**. Counterfactual error as we change the number of proxy variables, $S$, and the latent dimensionality, $D_{\mathbf{z}}$. We plot mean and $95\,\%$ CI over 5 realizations, intervening on the 25th, 50th, and 75th percentile value of t.

Figure 13: ATE absolute error varying the number of available proxies $(S)$ and the dimensionality of the latent space $(D_{\mathbf{z}})$. Mean and $95\%$ confidence interval over 5 realizations and all interventions, made in percentiles 25, 50 and 75 of t. Oracle represents a causal normalizing flow that observes $\mathbf{z}$.

**Experimental setup.** We consider two synthetic SCMs, with linear and non-linear causal equations, that follow the causal graph $\mathcal{G}$ depicted in the inset figure, comprising two independent hidden confounders affecting every variable, and $S$ null proxies. We evaluate how well DeCaFlow estimates $p(\mathrm{y}\,|\,\mathrm{do}(\mathrm{t}))$ as we change the number of proxy variables, $S$, and the specified latent dimensionality, $D_{\mathbf{z}}$.

In addition, we show the equations that we have used for the ablation study. There exist two unobserved confounders, $\mathbf{z}_1$ and $\mathbf{z}_2$. Note that the proxies available in the nonlinear experiment are bounded or periodic, specially sigmoids and hyperbolic tangents saturate and $\max(0, \mathrm{x})$ loses all the information about the confounder for negative values and sines and cosines are periodic functions. In other words, the distributions $p(\mathbf{z} \mid \mathbf{n}_i)$ are not complete, we lose information about $\mathbf{z}$ when in the transformations to each $\mathbf{n}$. However, if we add more proxies of the confounders, the information that the proxies contain about the confounder is higher, and the causal effect of $\mathrm{x}_1$ on $\mathrm{x}_2$ becomes recoverable.

Fig. 11: $\mathcal{G}$.

Linear

$$
\begin{cases}
\mathbf{z}_1 \sim P_{\mathbf{z}_1} \\
\mathbf{z}_2 \sim P_{\mathbf{z}_2} \\
\mathrm{t} = 1.5 \cdot \mathbf{z}_1 + 0.5 \cdot \mathbf{z}_2 + 0.4 \cdot \mathbf{u}_\mathrm{t} \\
\mathrm{y} = -0.75 \cdot \mathbf{z}_1 + 0.6 \cdot \mathbf{z}_2 + 0.9 \cdot \mathrm{t} + 0.3 \cdot \mathbf{u}_\mathrm{y} \\
\mathbf{n}_1 = -0.5 \cdot \mathbf{z}_1 + 0.3 \cdot \mathbf{z}_2 + 0.5 \cdot \mathbf{u}_2 \\
\mathbf{n}_2 = 0.75 \cdot \mathbf{z}_1 - 0.4 \cdot \mathbf{z}_2 + 0.4 \cdot \mathbf{u}_2 \\
\mathbf{n}_3 = -0.85 \cdot \mathbf{z}_1 + 0.6 \cdot \mathbf{z}_2 + 0.6 \cdot \mathbf{u}_3 \\
\mathbf{n}_4 = 0.6 \cdot \mathbf{z}_1 + 0.6 \cdot \mathbf{z}_2 + 0.55 \cdot \mathbf{u}_4 \\
\mathbf{n}_5 = -0.8 \cdot \mathbf{z}_1 + 0.4 \cdot \mathbf{z}_2 + 0.4 \cdot \mathbf{u}_5 \\
\mathbf{n}_6 = 0.9 \cdot \mathbf{z}_1 - 0.7 \cdot \mathbf{z}_2 + 0.6 \cdot \mathbf{u}_6 \\
\mathbf{n}_7 = -0.72 \cdot \mathbf{z}_1 + 0.5 \cdot \mathbf{z}_2 + 0.56 \cdot \mathbf{u}_8 \\
\mathbf{n}_8 = 0.78 \cdot \mathbf{z}_1 + 0.4 \cdot \mathbf{z}_2 + 0.58 \cdot \mathbf{u}_8 \\
\mathbf{n}_9 = -0.55 \cdot \mathbf{z}_1 + 0.7 \cdot \mathbf{z}_2 + 0.6 \cdot \mathbf{u}_9 \\
\mathbf{n}_{10} = 0.88 \cdot \mathbf{z}_1 + 0.3 \cdot \mathbf{z}_2 + 0.4 \cdot \mathbf{u}_{10}
\end{cases}
$$

Nonlinear

$$
\begin{cases}
\mathbf{z}_1 \sim P_{\mathbf{z}_1} \\
\mathbf{z}_2 \sim P_{\mathbf{z}_2} \\
\mathrm{t} = \dfrac{\mathbf{z}_1^2}{4} \cdot \sin\left(\dfrac{\mathbf{z}_2}{2}\right) + \mathbf{z}_1 + 0.6 \cdot \mathbf{u}_\mathrm{t} \\
\mathrm{y} = \dfrac{\mathbf{z}_1 \cdot \mathrm{t}}{4} + 0.8 \cdot \mathbf{z}_2 + 0.5 \cdot \mathrm{t} + \mathrm{x}_1 \cdot \mathbf{u}_2 \cdot 0.3 + 0.2 \cdot \mathbf{u}_\mathrm{y} \\
\mathbf{n}_1 = 0.6 \cdot \mathbf{z}_1^2 + \left(\dfrac{\mathbf{z}_2}{4}\right)^3 + 0.3 \cdot \sin\left(\dfrac{\mathbf{z}_2}{2}\right) + 0.5 \cdot \mathbf{u}_1 \\
\mathbf{n}_2 = \sin\left(\dfrac{\mathbf{z}_1}{2}\right) + \cos\left(\dfrac{\mathbf{z}_2}{3}\right) + 0.4 \cdot \mathbf{u}_2 \\
\mathbf{n}_3 = \cos\left(\dfrac{\mathbf{z}_1}{2}\right) - \tanh\left(\dfrac{\mathbf{z}_2}{3}\right) + 0.6 \cdot \mathbf{u}_3 \\
\mathbf{n}_4 = \tanh\left(\dfrac{\mathbf{z}_1}{2}\right) + \sigma\left(\dfrac{\mathbf{z}_2}{2}\right) + 0.55 \cdot \mathbf{u}_4 \\
\mathbf{n}_5 = \sigma\left(\dfrac{\mathbf{z}_1}{2}\right) + \max(0, -\mathbf{z}_2) + 0.4 \cdot \mathbf{u}_5 \\
\mathbf{n}_6 = \max(0, \mathbf{z}_1) - 0.5 \cdot \max(0, \mathbf{z}_2) + 0.6 \cdot \mathbf{u}_6 \\
\mathbf{n}_7 = \max(0, -\mathbf{z}_1) + 0.3 \cdot \max(0, -\mathbf{z}_2) + 0.5 \cdot \mathbf{z}_1 \cdot \mathbf{u}_7 \\
\mathbf{n}_8 = 0.8 \cdot \max(0, \mathbf{z}_1) + 0.3 \cdot \max(0, \mathbf{z}_2) + 0.58 \cdot \mathbf{u}_8 \\
\mathbf{n}_9 = 0.75 \cdot \max(0, -\mathbf{z}_1) + 0.5 \cdot \max(0, \mathbf{z}_2) + 0.6 \cdot \mathbf{u}_9 \\
\mathbf{n}_{10} = 0.3 \cdot \mathbf{z}_1^3 + 0.5 \cdot |\mathbf{z}_2| + 0.4 \cdot \mathbf{u}_{10}
\end{cases}
$$

**Results.** Fig. 12 shows the counterfactual error for every considered case, where we clearly observe that adding proxies reduces the error, with a drastic change as we add the second proxy, corroborating the requisites of Prop. 4.1. Similarly, underestimating $D_{\mathbf{z}}$ increases error (especially under causal sufficiency, $D_{\mathbf{z}} = 0$) while overestimating it does not seem to have an effect. This indicates that, indeed, the entropy term in Eq. 4 prevents non-shared information from being modeled through $\mathbf{z}$, as discussed in §3.

Next, we present in Fig. 13 the ATE error committed for each combination of proxies and latent dimension, complementing Fig. 12. If we observe the ATE error, we extract the same conclusion as observing counterfactual error, the causal effect is not recoverable with less than two proxies, and more proxies result in better estimates. On the other hand, the selection of

the dimension of the latent space bigger than the true dimension of the latent confounders does not affect the performance negatively.

## B.2 ABLATION STUDY FOR ENCODER SELECTION

We have performed an ablation study for selecting the encoder in the Sachs' dataset, where we evaluate the errors in the estimations of causal queries using a conditional normalizing flow (Flow) and a multilayer perceptron (MLP) as encoders. We also evaluate the impact of using the warm-up regularization [63] in the KL term. We can observe in Fig. 14 that we achieve lower errors when applying a regularized flow. This is able to model dependent latent variables and provides a more flexible representation. In addition, we can appreciate that applying the warm-up regularization term is useful and does not produce negative effects.

The improvement achieved by the flow is explained by the following practical aspects of the conditional normalizing flows. First, we can efficiently introduce the factorization proposed in Eq. 3, taking advantage of the structure of the

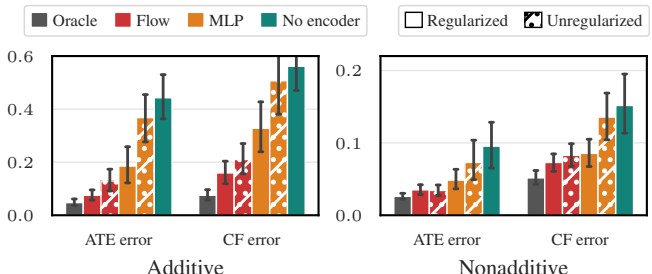

Figure 14: Ablation for encoder selection in Sachs' dataset. Metrics and $95\%$ CI over 5 realization and all confounded identifiable effects, intervening on percentiles 25, 50 and 75 of each intervened variable. Oracle represents a causal normalizing flow that observes all the confounders.

causal graph (see Fig. 23 for an example), while this factorization implies the use of several MLP. Second, normalizing flows are universal density approximators and do not need to assume specific posterior distributions (i.e. Gaussians). Note that every continuous distribution can be modeled by a conditional normalizing flow, following the Knöthe-Rosenblatt transport.

## B.3 ABLATION ON ENCODER FACTORIZATION

Using a conditional normalizing flow as the encoder allows us to model the dependencies between the observations and the posterior of the latent variables as desired.

We propose in Eq. 3 (extended in Eq. 60) a factorization in which each hidden confounder depends on its parents (other hidden confounders), its children and the parents of its children, avoiding cycles. If a child of an unobserved confounder, $c$, has other parents, then that child is a collider between the hidden confounders and the other parents of $c$. Therefore, conditioned on $c$, the hidden confounder is dependent of the other parents of $c$, given $c$. That is the reason because we consider sensible to include the other parents of $c$ in the factorization of the hidden confounder, $\mathbf{z}$.

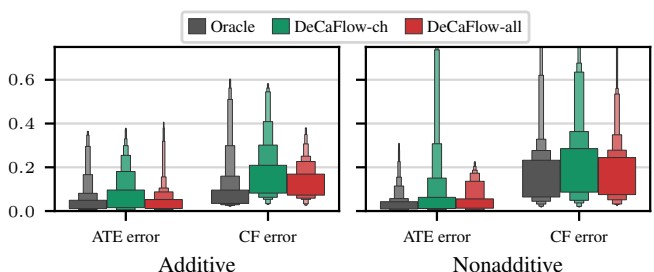

Figure 15: Ablation for posterior factorization in Ecoli dataset. Boxenplots of error metrics in the identifiable edges of Fig. 1. DeCaFlow-ch uses Eq. 57 and DeCaFlow-all uses Eq. 60 for posterior factorization.

However, we also provide an ablation study on the Ecoli dataset, where we show that this factorization indeed helps to the estimation of causal queries. Note that in the Ecoli dataset, `lacY` is a collider between `eutG` and `cspG`. Therefore, conditioned on `lacY`, the two hidden confounders `eutG` and `cspG` become dependent. The factorization of Eq. 60 implies that the posterior of `cspG` is modeled employing all the children of `cspG` and also the parents of its children, with `eutG` among them. This dependency can be modeled by our encoder in an autoregressive manner.

This factorization incorporates more variables to approximate the posterior of the hidden confounders, compared with a simpler approach that consist in modeling only children dependencies:

$$q_\phi(\mathbf{z} \mid \mathbf{x}) = \prod_{k=1}^{D_{\mathbf{z}}} q_\phi\left(\mathbf{z}_k \mid \mathrm{ch}(\mathbf{z}_k)\right) \tag{57}$$

As shown in Fig. 15, leveraging the factorization of Eq. 60 reduces the errors estimating causal queries in complex graphs, where colliders and dependent hidden confounders are present.

## B.4 SEMI-SYNTHETIC SACHS' DATASET

Table 1: Performance metrics on Sachs datasets. Mean$_{std}$ over five runs and all causal queries of interest. Interventions on `Raf`, `Mek` and `Akt` and evaluating on confounded identifiable effects. Bold indicates significantly better results (95% CI from a Mann-Whitney U test). Lower error values indicate better performance.

| | Model | Additive | | | | Nonadditive | | | |
|---|---|---|---|---|---|---|---|---|---|
| | | MMD obs $\times 10^4$ | MMD int $\times 10^4$ | \|ATE err\| $\times 10^2$ | \|CF err\| $\times 10^2$ | MMD obs $\times 10^4$ | MMD int $\times 10^4$ | \|ATE err\| $\times 10^2$ | \|CF err\| $\times 10^2$ |
| Oracle | CNF | $4.84_{1.84}$ | $7.50_{6.17}$ | $6.05_{6.83}$ | $10.03_{10.29}$ | $5.96_{2.37}$ | $6.71_{2.97}$ | $2.34_{2.02}$ | $4.84_{3.43}$ |
| Aware | DeCaFlow | $\mathbf{2.15_{0.54}}$ | $7.04_{3.87}$ | $\mathbf{4.49_{6.76}}$ | $\mathbf{12.95_{8.00}}$ | $5.12_{2.42}$ | $7.58_{16.92}$ | $\mathbf{5.16_{5.61}}$ | $1.83_{1.65}$ |
| | Deconfounder | – | – | $34.34_{33.45}$ | $71.13_{86.98}$ | – | – | $8.14_{10.69}$ | $63.15_{79.12}$ |
| | CNF | $5.80_{1.58}$ | $73.94_{88.78}$ | $44.49_{39.12}$ | $56.09_{38.89}$ | $5.11_{1.90}$ | $12.79_{20.73}$ | $9.74_{15.71}$ | $15.15_{15.37}$ |
| Unaware | ANM | $83.86_{13.41}$ | $110.28_{112.43}$ | $22.42_{14.06}$ | $29.40_{12.22}$ | $81.90_{7.21}$ | $60.40_{144.08}$ | $23.88_{13.94}$ | $28.97_{12.44}$ |
| | DCM | $87.80_{2.95}$ | $125.59_{118.20}$ | $21.21_{11.34}$ | $28.25_{6.96}$ | $14.23_{4.57}$ | $69.74_{390.81}$ | $8.44_{7.96}$ | $27.50_{23.71}$ |

This dataset represents a network of protein-signaling in human T lymphocites. Every variable, except `PKA` and `Plcg` can be intervened upon; therefore, there is not only one causal query of interest, but tens of possible causal queries can arise in this setting. This highlights one of the strengths of DeCaFlow, because we only need a single trained model to answer all identifiable causal queries.

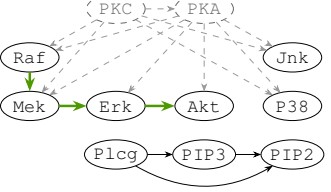

Figure 16: Sachs' graph. Green edges mark proxy-identifiable effects.

The original data contains a total of 853 observational samples; however, we have decided to evaluate our model on semi-synthetic data because of the following reasons:

- The original network of Sachs et al. [53] contains cycles, which is a violation of one of our assumptions. However, we have found different versions of the causal graph [28, 39] that do not contain cycles. Therefore, we have decided to employ the causal graph that appears in the library *bnlearn* [57]—a recognized library for Bayesian Nerwork learning—as ground truth causal graph. The best way to ensure that the causal graph used is the ground truth is by generating samples according to the causal graph. In addition, that causal graph is the one used by Chao et al. [9].
- We can compare our model with one of the baseline models, DCM, with the same dataset as Chao et al. [9] used.
- Semi-synthetic data allow us to compute all metrics to evaluate causal queries, having the ground truth.

For generating the data in this experiment, we have followed the procedure proposed by Chao et al. [9], where they take the causal graph of Sachs et al. [53] and the empirical distribution of the root nodes, and generate the rest of the variables with random non-linear mechanisms. In addition, exogenous variables have been included in an additive and non-additive manner, respectively.

In the following, we complement the figures presented in §5 with a table that summarizes all the interesting metrics, evaluated on the confounded identifiable causal queries shown in Fig. 16. Interventional distributions and counterfactuals have been computed intervening in percentiles 25, 50 and 75 of the intervened variable.

Since observational MMD is computed only once, the statistics given in Tab 1 are calculated *only* over 5 runs. On the other hand, we have as many interventional MMDs per run as interventions have been made. However, the statistics of interventional MMD are computed over all the interventions of all intervened variables and 5 runs (5 runs × 3 intervened variables = 15 samples). Finally, statistics over counterfactual error and ate error aggregate all the intervention-outcome pairs over the five runs. For example, in this case we intervene in 3 variables, performing 3 different interventions and evaluate in 3, 2, and 1 variable, respectively, for each intervened variable, and we have a total of (3+2+1)×3×5 = 90 different measurements to compute the statistics.

The metrics in Tab 1 indicate that DeCaFlow outperforms all baselines across all interventional and counterfactual causal queries in both settings of the semi-synthetic datasets. However, as discussed in §6, a limitation of our empirical approach is that the differences in observational MMD, the selection criterion for CGMs, are marginal between the *oracle*, DeCaFlow, and CNF. Notably, DeCaFlow even achieves a lower MMD than the *oracle*. This discrepancy arises because the number of variables is large, and the MMD differences are on the order of $10^{-4}$.

## B.5 SEMI-SYNTHETIC ECOLI70 DATASET

The Ecoli 70 dataset represent the gene expression of 46 genes of the RNA-seq of *Escherichia coli* bacteria. The assumed causal graph comes from the study of [56], which provides insight into the regulatory mechanisms governing *E. coli* gene expression. Examples of interventions in these networks are gene knockout and gene overexpression [37]. A priori, there could be several variables in which intervening can be interesting in evaluating the effects in the cell.

Table 2: Performance metrics on Ecoli70 dataset. ATE and CF error statistics computed aggregating all causal queries and 5 runs. Intervened and evaluated on the direct confounded identifiable causal effects of Fig. 1. Bold indicates significantly better results (95% CI from a Mann-Whitney U test). Lower error values indicate better performance.

| | Model | Additive | | | | Nonadditive | | | |
|---|---|---|---|---|---|---|---|---|---|
| | | MMD obs $\times 10^4$ | MMD int $\times 10^4$ | \|ATE err\| $\times 10^2$ | \|CF err\| $\times 10^2$ | MMD obs $\times 10^4$ | MMD int $\times 10^4$ | \|ATE err\| $\times 10^2$ | \|CF err\| $\times 10^2$ |
| Oracle | CNF | $2.34_{0.62}$ | $6.05_{5.28}$ | $5.04_{7.42}$ | $9.91_{12.46}$ | $1.49_{0.57}$ | $4.05_{8.22}$ | $3.51_{4.84}$ | $1.67_{1.64}$ |
| Aware | DeCaFlow | $2.42_{0.82}$ | $\mathbf{7.04_{3.87}}$ | $\mathbf{4.49_{6.76}}$ | $\mathbf{12.95_{8.00}}$ | $1.58_{0.65}$ | $9.22_{22.38}$ | $\mathbf{8.79_{17.91}}$ | $2.15_{2.10}$ |
| | Deconfounder | – | – | $27.35_{26.17}$ | $82.15_{116.90}$ | – | – | $30.00_{33.24}$ | $9.90_{9.47}$ |
| | CNF | $2.98_{1.15}$ | $10.25_{12.13}$ | $23.91_{25.16}$ | $34.02_{23.90}$ | $1.95_{0.77}$ | $10.20_{20.87}$ | $12.72_{19.21}$ | $2.45_{2.06}$ |
| Unaware | ANM | $32.80_{2.81}$ | $44.33_{17.62}$ | $21.88_{23.89}$ | $31.33_{20.64}$ | $13.17_{3.95}$ | $27.56_{31.57}$ | $15.04_{18.18}$ | $2.71_{1.88}$ |
| | DCM | $31.65_{0.27}$ | $49.50_{36.83}$ | $24.45_{33.31}$ | $30.22_{24.83}$ | $18.78_{6.01}$ | $33.37_{36.14}$ | $15.07_{22.37}$ | $2.36_{2.08}$ |

For this experiment, we have generated the data in the same way as done with Sachs' dataset with random mechanisms, but in this case, since we do not have enough samples, root nodes follow standard Gaussian distributions. We have included an additive and a nonadditive ways of including exogenous variables. In this case, we have used a semi-synthetic dataset because the real dataset available in *bnlearn* [57] contains only 9 samples.

In Fig. 1 we show the causal graph of this setting. In addition, note that Fig. 1 has been extracted from our Alg. 6 of causal effect identfiability. That is, we have specified the causal graph and the variables that are unmeasured, and our Algorithm returns (in green) all the paths that are identifiable by DeCaFlow. Consider that black arrows are also identifiable, not only by DeCaFlow, but also for any CGM that approximates the observed data. In red, arrows that are not identifiable by DeCaFlow because there are not enough proxies to infer an unbiased causal effect.

A table summarizing the results obtained in the estimation confounded identifiable causal queries are presented in Tab 2. The statistics have been computed in the same way as in Sachs' dataset. In the case of ATE and CF error, they have been computed only on the *direct* confounded identifiable paths, i.e., the green paths in Fig. 1.

DeCaFlow significantly outperforms the baselines in ATE and counterfactual estimation in the additive setting and in ATE estimation in the nonadditive setting. The MMD differences, both observational and interventional, are negligible between the *oracle*, DeCaFlow, and CNF, likely due to the high number of variables diluting estimation bias. Counterfactual differences in the nonadditive setting are also insignificant. However, compared to the *oracle*, the gap between the *oracle* and *unaware* CGMs is smaller than in the additive case. While DeCaFlow reaches an intermediate point, the difference remains insignificant.

### B.5.1 Comment on Deconfounder results

One may realize that the errors committed by the Deconfounder of [65, 66] are greater than the errors committed by the unaware models. First of all, we want to underline that, although the Deconfounder allows us to predict counterfactuals, the algorithm does not present any guarantees of a correct counterfactual estimation because it does not model the exogenous variables of the SCM. That is the reason of the bad performance in couterfactual estimation.

On the other hand, let us justify some of the other paths where the errors of the Deconfounder are greater than unaware models. In Sachs' datasetto model the causal effect Ekt→Akt, the factorization model of the deconfounder uses Raf, Mek, Jnk and P38 to extract the substitute confounder; the factorization model assumes that all those variables are independent conditioned to $\tilde{z}$, while that is not the case in the true SCM and, therefore, this SCM violates the independence assumption of [65]. The same argument is valid for the paths yceP→yfaD, lacA→yaeM, yceP→yfaD, ydeE→pspA and pspB→pspA.

On the other hand, the paths lacZ→yaeM, asnA→lacY are frontdoor paths that DeCaFlow can identify because it models the hidden confounder following the true causal graph. However, the Deconfounder is not designed to model this paths. To evaluate its performance for frontdoor paths, Deconfounder uses the same variables as DeCaFlow to extract the substitute of the confounder. However, the Deconfounder assumes independence conditioned to the substitute confounder and that is not the case; therefore, we are violating the independence assumption again.

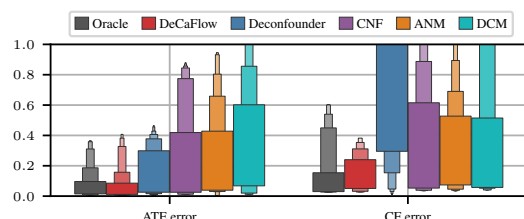

Figure 17: ATE and CF error evaluating only links where deconfounder should work in the additive case.

Table 3: Performance metrics on Ecoli70 dataset. Statistics computed an all samples over 5 runs, intervening and evaluating only in the causal effects that Deconfounder should solve. Bold indicates significantly better results (95% CI from a Mann-Whitney U test). Lower error values indicate better performance.

| | Model | $|\text{ATE err}| \times 10^2$ | $|\text{CF err}| \times 10^1$ |
|---|---|---|---|
| Oracle | CNF | $8.31_{10.95}$ | $1.49_{1.86}$ |
| Aware | DeCaFlow | $\mathbf{7.78_{7.30}}$ | $\mathbf{1.87_{1.50}}$ |
| | Deconfounder | $14.35_{15.24}$ | $12.03_{15.81}$ |
| Unaware | CNF | $27.82_{30.17}$ | $4.01_{3.62}$ |
| | ANM | $27.63_{29.74}$ | $3.64_{3.15}$ |
| | DCM | $42.45_{54.23}$ | $4.08_{4.12}$ |

Table 4: Performance metrics on Ecoli70 dataset. Statistics computed on all *unconfounded* direct effects and 5 runs. Bold indicates significantly better results (95% CI from a Mann-Whitney U test). Lower error values indicate better performance.

| | Model | Additive | | | Nonadditive | | |
|---|---|---|---|---|---|---|---|
| | | MMD int $\times 10^4$ | $|\text{ATE err}| \times 10^2$ | $|\text{CF err}| \times 10^2$ | MMD int $\times 10^4$ | $|\text{ATE err}| \times 10^2$ | $|\text{CF err}| \times 10^2$ |
| Oracle | CNF | $3.72_{3.73}$ | $2.00_{2.27}$ | $1.27_{3.49}$ | $1.94_{2.96}$ | $1.92_{1.99}$ | $1.76_{4.10}$ |
| Aware | DeCaFlow | $4.53_{4.98}$ | $2.00_{2.07}$ | $1.31_{2.93}$ | $2.83_{6.36}$ | $1.93_{1.95}$ | $1.62_{3.87}$ |
| Unaware | CNF | $4.77_{6.09}$ | $2.02_{2.21}$ | $1.22_{3.18}$ | $2.97_{7.64}$ | $1.95_{1.92}$ | $1.71_{3.93}$ |
| | ANM | $34.72_{8.56}$ | $3.57_{3.02}$ | $2.02_{4.09}$ | $15.13_{12.57}$ | $3.53_{3.15}$ | $2.64_{5.34}$ |
| | DCM | $36.23_{14.29}$ | $3.48_{2.75}$ | $2.69_{2.30}$ | $21.22_{13.68}$ | $3.42_{2.63}$ | $3.00_{3.42}$ |

The only two paths that meet the Deconfounder assumptions in Fig. 1 are lacA→lacY and yedE→pspB. And we can observe that in those paths, the Deconfounder performs at least as well as unaware methods. On the other hand, all the factor models used for the Deconfounder implementation (PPCA, Deep exponential families and Variational autoencoder) assume additive noise. Therefore, interventional distributions in nonadditive settings are not computable theoretically with these models.

### B.5.2 Metrics on the other paths

In this subsection we include a comparison between all the models in the *unconfounded* and the unidentifiable effects. For *unconfounded effects*, our expectation is to observe that all the CGMs achieve a performance comparable with the *oracle*. On the other hand, we expect to have higher errors in unidentifiable effects, since we do not have theoretical guarantees.

**Unconfounded Effects.** The results for *unconfounded effects* are summarized in Fig. 18 and Tab 4, considering only direct effects for ATE and counterfactual error computations. As expected, DeCaFlow and CNF achieve metrics comparable to the *oracle* in both ATE and counterfactual estimations, particularly evident in Fig. 18, where error distributions are nearly identical. 4 does not show

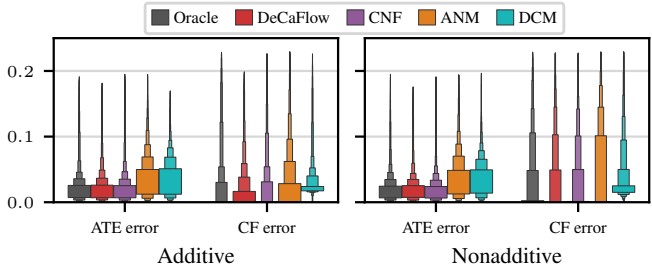

Figure 18: Error boxplots on the Ecoli70 dataset for different CGMs, averaged over all *unconfounded* direct effects (see Fig. 1) after intervening in their 25th, 50th, and 75th percentiles and 5 random realizations of the experiment.

statistically significative differences between DeCaFlow and CNF. Notably, architectures based on causal normalizing flows outperform ANM and DCM, which model each causal mechanism, $f_i$, with separate networks. This difference is crucial in settings with many variables and complex relations, where scalability is essential. Unlike ANM and DCM, which suffer from error propagation and limited scalability, causal normalizing flows leverage a single amortized model, making them more efficient in high-dimensional scenarios.

Finally, note that the Deconfounder has not been included in these metrics because it is not designed for *unconfounded queries* and there are many queries, while one Deconfounder model is needed for each query.

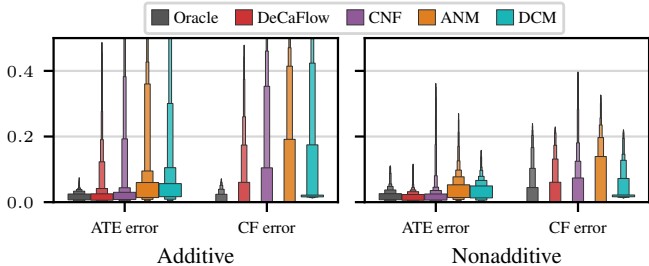

Figure 19: Error boxenplots on the Ecoli70 dataset for different CGMs, averaged over all unidentifiable direct effects (see Fig. 1) after intervening in their 25th, 50th, and 75th percentiles and 5 random realizations of the experiment.

Table 5: Performance metrics on Ecoli70 dataset. Statistics computed on all unidentifiable direct effects and 5 runs. Bold indicates significantly better results (95% CI from a Mann-Whitney U test). Lower error values indicate better performance

| | Model | Additive | | | Nonadditive | | |
|---|---|---|---|---|---|---|---|
| | | MMD int $\times 10^4$ | \|ATE err\| $\times 10^2$ | \|CF err\| $\times 10^3$ | MMD int $\times 10^5$ | \|ATE err\| $\times 10^2$ | \|CF err\| $\times 10^2$ |
| Oracle | CNF | $3.71_{3.52}$ | $1.79_{1.36}$ | $5.88_{15.16}$ | $16.98_{6.87}$ | $1.75_{1.59}$ | $1.62_{4.57}$ |
| Aware | DeCaFlow | $3.80_{3.61}$ | $3.95_{7.89}$ | $33.62_{80.37}$ | $23.02_{21.96}$ | $1.75_{1.66}$ | $1.88_{4.97}$ |
| Unaware | CNF | $4.54_{4.81}$ | $4.75_{10.65}$ | $44.76_{126.36}$ | $20.22_{6.68}$ | $2.32_{3.80}$ | $2.13_{6.25}$ |
| | ANM | $34.38_{5.17}$ | $7.43_{12.64}$ | $52.70_{137.99}$ | $130.71_{41.64}$ | $4.01_{3.82}$ | $2.93_{7.21}$ |
| | DCM | $35.49_{4.95}$ | $7.67_{13.93}$ | $67.46_{132.21}$ | $198.23_{58.62}$ | $3.43_{2.76}$ | $3.29_{3.92}$ |

**Unidentifiable Effects.** The results for unidentifiable effects—causal queries that violate the assumptions in §4—are summarized in Fig. 19 and Tab 5. Notably, the *oracle* performs significantly better than the other CGMs. As seen in Fig. 19, error distributions are highly skewed, with ATE and counterfactual errors reaching extreme values—considering that metrics are computed on the standardized variables. Tab 5 shows no significant differences between the metrics achieved by DeCaFlow and CNF.

### B.5.3 Hyper-parameters and splits

We have performed a hyperparameter grid search in both experiments on semi-synthetic datasets, exploring a large combination of hyperparameters for each model and dataset.

These are the parameters that were modified for each model:

- CNF: the number of neurons and hidden layers of the single-layer flow, the type of flow (MAF, NSF). LR scheduler reducing on plateau and early stopping were applied with Adam optimizer [30].
- DeCaFlow: number of neurons and hidden layers of the single-layer causal flow (generative network), type of flow of generative network (MAF, NSF), number of neurons and hidden layers of the single-layer encoder flow (inference network), type of encoder flow (MAF, NSF), KL regularization (True, False). LR scheduler reducing on plateau and early stopping was applied with the Adam optimizer [30].
- Deconfounder: type of factorization model (PPCA, VAE, Deep Exponential Families), number of neurons and hidden layers (in case of deep models), type of outcome model (MLP, random forest, linear regression), number of neurons and hidden layers of the outcome model (in case of deep models).
- DCM: number of neurons and hidden layers of each network, learning rate and number of iterations (we have not introduced early stopping or learning rate scheduler). The rest of hyperparameters were selected to the default value in the original code.
- ANM: an automatic search was performed across several models in the original DCM code. This search is performed with the DoWhy package [7].

The selection was based on the matching of the observational for the causal generative models and, in the Deconfounder, the factorization networks were selected by the likelihood of the observed variables and the outcome models with maximum likelihood.

| Model | Epoch Tr. [s] (20000 samples) | Interventional [s] (2500 samples) | CF [s] (2500 samples) |
|---|---|---|---|
| Oracle | $0.64_{0.06}$ | $0.30_{0.02}$ | $0.36_{0.03}$ |
| DeCaFlow | $0.98_{0.10}$ | $0.28_{0.02}$ | $0.35_{0.04}$ |
| CNF | $0.60_{0.07}$ | $0.26_{0.01}$ | $0.32_{0.05}$ |

Table 6: Computation times per model across training and evaluation regimes for Ecoli Additive Dataset. Mean and standard deviation of the training and inference time over 100 epochs in training and over 7 interventions in inference.

| Model | Epoch Tr. [s] (20000 samples) | Interventional [s] (2500 samples) | CF [s] (2500 samples) |
|---|---|---|---|
| Oracle | $0.32_{0.06}$ | $0.08_{0.001}$ | $0.102_{0.010}$ |
| DeCaFlow | $0.75_{0.12}$ | $0.05_{0.004}$ | $0.086_{0.005}$ |
| CNF | $0.33_{0.06}$ | $0.048_{0.003}$ | $0.065_{0.006}$ |

Table 7: Computation times on the Sachs' Additive Dataset. Mean and standard deviation of the training and inference time over 100 epochs in training and over 3 interventions in inference.

Although including all hyperparameters would be very extensive, we give here a sample of the hyperparameters selected for DeCaFlow in the Ecoli additive dataset:

- Hidden neurons of causal flow (generative network): $3 \times 128$
- Type of causal flow (generative network): neural spline flow (NSF) [13].
- Hidden neurons of encoder flow (inference network): $3 \times 64$
- Type of flow (inference network): neural spline flow (NSF) [13].
- Regularize: True (warm-up: 30 epochs)
- Total number of parameters: 182k.

Both experiments were performed with 25,000 data, split into $80\%, 10\%, 10\%$ (train, validation, and test). All metrics are given over the test dataset.

### B.5.4 Processing times

All the experiments were conducted on CPU. Although the experiments were carried out on a cluster of different CPU, we include here two tables for the two semi-synthetic datasets (Tab 6 and Tab 7) with the processing times measured in a CPU Intel(R) Core(TM) i7-13650HX laptop, just to show that even in a laptop CPU, the training and inference times are sensible even for large datasets as the Ecoli dataset.

Note that DeCaFlow takes more time in training. This is because the network is more complex, due to the inference network, and that we have to sample from the posterior distribution. However, the difference in inference is not that relevant. In fact, DeCaFlow takes less time than the oracle in inference, even when they are sampling the same number of variables (hidden confounders + observed variables). The unaware causal flow (CNF) only samples from the observed variables. That is why the inference time is lower.

### B.6 LAW SCHOOL FAIRNESS USE-CASE

Taking inspiration from the experiments by Kusner et al. [36] and Javaloy et al. [25] we test whether, by modeling the confounded SCM with DeCaFlow, we can leverage it for more than causal-query estimation and, in particular, for counterfactual-fairness prediction.

**Dataset and objective.** Our aim is to train a gradient-boosted decision tree [16] on the law school dataset [68], which comprises of 21 790

law students who were admitted by the Law School Admissions Council (LSAC) from 1991 through 1997. We have performed an experiment similar to that carried out by Kusner et al. [36], where race and sex were treated as sensitive attributes. We have considered the following variables to include in our study:

- `Race`: binary indicator of the race that distinguish between white and non-white.
- `Sex`: binary indicator of the sex that distinguish between male and female.
- `Fam`: family income.

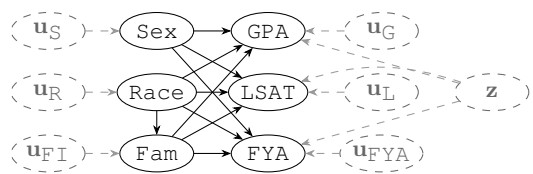

Figure 21: Confounded SCM modeled by DeCaFlow.

- `LSAT`: the grade achieved in the Law School Admission Test (LSAT).
- `UGPA`: the undergraduate grade point average (GPA) of the student previous to the admission.
- `FYA`: first-year average grade.
- `Decile3`: the decile of the grades in the third year of university. This is the variable to predict.

For our purpose, we consider that an estimator, $\hat{y}$, is fair if it meets *Demographic parity*, defined in [36, Def. 3] as follows. A predictor $\hat{y}$ satisfies demographic parity if the predicted distributions for different values of a sensitive attribute are equal: $p(\hat{y} \mid t = 0) = p(\hat{y} \mid t = 1)$. We evaluate the difference between predicted distributions using MMD—a lower distance between the predictions for the two groups of a sensitive attributes denotes a fairer predictor.

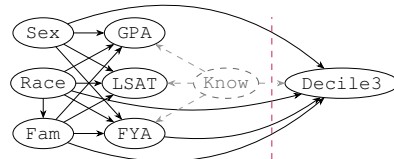

Figure 20: Assumed causal graph in §B.6. Only the classifiers consider `Decile3`.

The assumed causal graph is slightly different from that of Kusner et al. [36], since their purpose is to make a fair prediction `FYA` accounting only for `Race`, `Sex`, `LSAT` and `UGPA`. However, we include `Fam` and `FYA` as predictors and the task is to predict `Decile3` and the assumed causal graph is the one of Fig. 20.

**Proposed DeCaFlow-based fair predictor.** We propose to model the confounded causal graph presented in Fig. 21, where are explicitly shown the exogenous variables, that are independent of the other variables of the graph except of their associated endogenous variable.

Afterwards, we predict the outcome, `Decile3` from the extracted latent variable that acts as substitute of the `knowledge` and the exogenous variables of `FYA` and `Fam`, following the causal graph of Fig. 20, using a gradient-boosted decision tree [16]: $\tilde{p}(\texttt{Decile3} \mid \mathbf{u}_{FI}, \mathbf{u}_{FYA}, \mathbf{z})$. DeCaFlow models $\mathbf{z}$ and the exogenous variables as independent from `Race` and `Sex`. Therefore, the prediction of `Decile3` should be we more fair yet slightly less accurate.

**Baselines.** The baselines used to compare our approach are the methods *Fair K* and *Fair add* proposed in Kusner et al. [36]. *Fair K* is a fair predictor categorized in Level 2 in Kusner et al. [36], which postulates that the student's knowledge, `know` affects `GPA`, `LSAT`, `FYA` and `Decile 3`, following the distributions described below.

$$
\begin{aligned}
\texttt{Fam} &\sim \mathcal{N}\left(b_{Fam} + w_{Fam}^{R}\texttt{Race}, 1\right), \\
\texttt{GPA} &\sim \mathcal{N}\left(b_{G} + w_{G}^{K}\texttt{know} + w_{G}^{R}\texttt{Race} + w_{G}^{S}\texttt{Sex} + w_{G}^{Fam}\texttt{Fam}, \sigma_{G}^{2}\right), \\
\texttt{LSAT} &\sim \text{Poisson}\left(\exp(b_{L} + w_{L}^{K}\texttt{know} + w_{L}^{R}\texttt{Race} + w_{L}^{S}\texttt{Sex} + w_{L}^{Fam}\texttt{Fam})\right), \\
\texttt{FYA} &\sim \mathcal{N}\left(w_{F}^{K}\texttt{know} + w_{F}^{R}\texttt{Race} + w_{F}^{S}\texttt{Sex} + w_{F}^{Fam}\texttt{Fam}, 1\right), \\
\texttt{Decile3} &\sim \text{Poisson}\left(\exp(w_{D}^{K}\texttt{know} + w_{D}^{R}\texttt{Race} + w_{D}^{S}\texttt{Sex} + w_{D}^{Fam}\texttt{Fam})\right), \\
\texttt{know} &\sim \mathcal{N}(0, 1).
\end{aligned}
\tag{58}
$$

Then, the posterior distribution `know` is inferred using Monte Carlo with the probabilistic programming language Pyro [6]. The outcome is predicted using the inferred `know` using a gradient-boosted decision tree [16]: $\tilde{p}(\texttt{Decile3} \mid \texttt{know})$.

On the other hand, *Fair Add* predicts the outcome from the residuals of predicting each variable with each parent, which guarantees that these residuals are independents of `Race` and `Sex`. That is, the predictor estimates the distribution $p(\texttt{Decile3} \mid \mathbf{r}_{Fam}, \mathbf{r}_{UGPA}, \mathbf{r}_{LSAT}, \mathbf{r}_{FYA})$, where these residuals are computed as:

$$
\begin{aligned}
\mathbf{r}_{Fam} &= \texttt{Fam} - \mathbb{E}\left[\texttt{Fam} \mid \texttt{Sex}, \texttt{Race}\right] \\
\mathbf{r}_{UGPA} &= \texttt{UGPA} - \mathbb{E}\left[\texttt{GPA} \mid \texttt{Sex}, \texttt{Race}, \texttt{Fam}\right] \\
\mathbf{r}_{LSAT} &= \texttt{LSAT} - \mathbb{E}\left[\texttt{LSAT} \mid \texttt{Sex}, \texttt{Race}, \texttt{Fam}\right] \\
\mathbf{r}_{FYA} &= \texttt{FYA} - \mathbb{E}\left[\texttt{FYA} \mid \texttt{Sex}, \texttt{Race}, \texttt{Fam}\right]
\end{aligned}
\tag{59}
$$

All predictors used are gradient-boosted decision trees [16].

**Results.** Tab 8 provides the prediction error (RMSE) and the difference between group distributions (MMD) for the proposed DeCaFlow-based predictor, comparing with an Unfair predictor that uses sensitive attributes; an Unaware predictor that excludes sensitive attributes, and two fair predictors, Fair K and Fair Add, as initially proposed by Kusner et al. [36].

We observe in Fig. 22 that DeCaFlow yields a much fairer predictor than the Unfair one, as the per-race predicted distributions remain much closer together. Looking at Tab 8, we find that this indeed comes at a small cost in RMSE, corroborating our intuitions, while the other two fair predictors incur a much higher predictive cost.

More in detail, although the *fair* methods proposed by Kusner et al. [36] achieve significantly better *demographic parity* than our approach using DeCaFlow (as indicated by a much lower MMD), their predictive performance is substantially inferior. Specifically, their performance is comparable to predicting the outcome using only the mean of the distribution, which serves as a baseline in Tab 8. In contrast, DeCaFlow achieves a 98% reduction in MMD while incurring only an 11% increase in RMSE, as illustrated in Fig. 22.

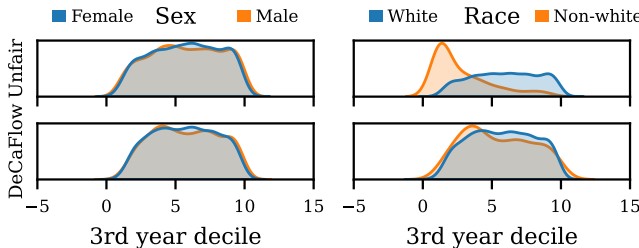

Figure 22: **Distribution of predicted `Decile3`**. A fair predictor yields similar distributions across the considered groups per attribute (`Sex` and `Race`).

Table 8: Test RMSE on `Decile3` prediction and MMD of inter-group predictive distributions.

|  | Unfair | Unaware | DeCaFlow | Fair $K$ | Fair Add | Mean |
|---|---|---|---|---|---|---|
| RMSE | 1.413 | 1.419 | 1.604 | 2.817 | 2.826 | 2.83 |
| MMD | 0.163 | 0.147 | 0.0054 | $10^{-5}$ | $10^{-4}$ | 0 |

These experiments demonstrate that leveraging DeCaFlow to model confounded Structural Causal Models is beneficial beyond causal query estimation, leading to improved overall performance.

# C IMPLEMENTATION DETAILS

## C.1 POSTERIOR FACTORIZATION OF THE DECONFOUNDING NETWORK

DeCaFlow is capable of modeling confounded SCMs that contain several hidden confounders, $\mathbf{z} = \{\mathbf{z}_k\}_{k=1}^{D_\mathbf{z}}$, as in the Sachs' dataset (Fig. 16), Ecoli dataset (Fig. 1) or the Napkin graph (Fig. 9). In such cases, the posterior over latent variables factorizes. We propose a factorized posterior in which each hidden confounder is conditioned on its children and the parents of its children.

$$q_\phi(\mathbf{z} \mid \mathbf{x}) = \prod_{k=1}^{D_\mathbf{z}} q_\phi \left( \mathbf{z}_k \mid \mathrm{pa}(\mathbf{z}_k) \cup \mathrm{ch}(\mathbf{z}_k) \cup \bigcup_{c \in \mathrm{ch}(\mathbf{z}_k)} (\mathrm{pa}(c) \setminus \{\mathbf{z}_j : j \geq k\}) \right) \tag{60}$$

Since we propose to use a conditional normalizing flow as the encoder, the interdependencies between the hidden confounders are modeled in an autoregressive manner.

The rightmost part of the conditioning set accounts for collider-induced associations: conditioning on a child of $\mathbf{z}_k$, $c$, makes $\mathbf{z}_k$ dependent on other parents of $c$. Other parents of $c$ can also be hidden confounders. To model this, a causal ordering of the $\mathbf{z}$ components is assumed to avoid cycles in factorization, but it does not affect estimation, as collider associations have no inherent causal direction.

## C.2 REGULARIZATION OF THE KULLBACK-LEIBLER TERM IN ELBO

We propose the implementation of a warm-up adaptive regularization term that weights the contribution of the Kullback-Leibler term in the ELBO, to avoid posterior collapse [63].

In the training loop, if the current epoch is lower than the predefined warm-up parameter, the KL term is weighted by $\beta$, that is defined as $\beta = \min(1, \mathrm{KL}[q_\phi(\mathbf{z} \mid \mathbf{x}) \| p(\mathbf{z})])$.

**Algorithm 1** KL regularization term in the training loop

---

1: **function** ELBO COMPUTATION(epoch, warmup, $\theta, \phi$)
2:    **if** epoch $<$ warmup:
3:       $\mathcal{L}(\phi, \theta) = \mathbb{E}_{q_\phi}[\log p_{\boldsymbol{\theta}}(\mathbf{x}|\mathbf{z})] - \beta \cdot \text{KL}[q_\phi(\mathbf{z}|\mathbf{x})\|\, p(\mathbf{z})]$
4:    **else**:
5:       $\mathcal{L}(\phi, \theta) = \mathbb{E}_{q_\phi}[\log p_{\boldsymbol{\theta}}(\mathbf{x}|\mathbf{z})] - \text{KL}[q_\phi(\mathbf{z}|\mathbf{x})\|\, p(\mathbf{z})]$
6:    **return** $\mathcal{L}$
7: **end function**

---

With this, we encourage the model to improve the reconstruction of the data in the first epochs, ignoring the KL term if the posterior is very similar to the prior, i.e., if $KL \approx 0$, then $\beta \approx 0$ and $\mathcal{L}(\phi, \theta) \approx \mathbb{E}_{q_\phi}[\log p_{\boldsymbol{\theta}}(\mathbf{x}|\mathbf{z})]$. After the warm-up epoch, the loss is equivalent to the regular expression of ELBO.

We have tested in the ablation study of §B.2 that the inclusion of the regularization term is useful in the Sachs' dataset. On the other hand, when posterior collapse does not occur, the $\beta$ term will be upper bounded by 1, therefore, not affecting the training process.

## C.3 STRUCTURAL INDUCTIVE BIAS

As presented in the original paper of Javaloy et al. [25], the **adjacency matrix** that represents the causal graph is an input of the normalizing flow. In this case, we introduce the structural constrains between **i)** exogenous and endogenous variables and **ii)** conditional variables and endogenous variables.

This allows that our deconfounding network factorizes the posterior distribution as shown in Eq. 3, modeling each hidden confounder as a function of its children, its parents and the parents of its children.

On the other hand, the structural information in the generative network allow to model each endogenous variable exclusively from its parents, whether its parents are other endogenous variables or hidden confounders, following Eq. 2.

We include in Fig. 23 a fully-detailed illustration of the architecture of DeCaFlow for the Napkin causal graph (Fig. 9), where it is shown in detail how its structural constraint is introduced in each conditional normalizing flow. To do so, DeCaFlow exploits MADE (Masked Autoencoder for Distribution Estimation) [17], which can be implemented with custom masks that specify the functional dependencies [10].

Finally, note that the do-operator is inherited from the original paper of Causal Normalizing flows, and the details about it deserve a new section: §D.

# D DO-OPERATOR

We introduce here the algorithms that DeCaFlow employ to generate interventional samples and counterfactuals. But first, we include those of Javaloy et al. [25], since we leverage these CNFs as building blocks for DeCaFlow. Note that the notation applied for DeCaFlow is slightly different from the that used in the causal flows, naming the intervened variable as t, instead of $x_i$, in order to be consistent with the notation used in §2 and §4. However, note that both variables play the same role, and that t $\subset$ **x**.

## D.1 DO-OPERATOR IN CAUSAL NORMALIZING FLOWS

---

**Algorithm 2** Algorithm to sample from the interventional distribution, $P(\mathbf{x} \mid \text{do}(\mathbf{x}_i = \alpha))$. From Javaloy et al. [25].

---

1: **function** SAMPLEINTERVENEDDIST$(i, \alpha)$
2:    $\mathbf{u} \sim P_{\mathbf{u}}$                                          ▷ Sample a value from the observational distribution.
3:    $\mathbf{x} \leftarrow T_\theta^{-1}(\mathbf{u})$
4:    $\mathbf{x}_i \leftarrow \alpha$                                      ▷ Set $x_i$ to the intervened value $\alpha$.
5:    $\mathbf{u}_i \leftarrow T_\theta(\mathbf{x})_i$                                ▷ Change the $i$-th value of **u**.
6:    $\mathbf{x} \leftarrow T_\theta^{-1}(\mathbf{u})$
7:    **return x**                                       ▷ Return the intervened sample.
8: **end function**

---

The computation of counterfactuals follows the steps of *abduction, action and prediction* postulated by [48]. The *abduction* step consists of using the observations to determine the value of the exogenous variables. Then, *action* is computing the

intervention, modifiying the causal mechanism of the intervened variable and *prediction* consist of using the exogenous variables and the modified SCM to compute the counterfactual.

---

**Algorithm 3** Algorithm to sample from the counterfactual distribution, $P(\mathbf{x}^{\text{cf}} \mid \text{do}(\mathbf{x}_i = \alpha), \mathbf{x}^{\text{f}})$. From Javaloy et al. [25].

---

1: **function** GETCOUNTERFACTUAL($\mathbf{x}^{\text{f}}, i, \alpha$)
2:     $\mathbf{u} \leftarrow T_\theta(\mathbf{x}^{\text{f}})$                ▷**Abduction:** Get $\mathbf{u}$ from the factual sample.
3:     $\mathbf{x}_i^{\text{f}} \leftarrow \alpha$                ▷**Action:** Set $x_i$ to the intervened value $\alpha$.
4:     $\mathbf{u}_i \leftarrow T_\theta(\mathbf{x}^{\text{f}})_i$            ▷**Action:** Change the $i$-th value of $\mathbf{u}$.
5:     $\mathbf{x}^{\text{cf}} \leftarrow T_\theta^{-1}(\mathbf{u})$           ▷**Prediction:** Get counterfactual
6:     **return** $\mathbf{x}^{\text{cf}}$              ▷ Return the counterfactual value.
7: **end function**

## D.2   DO-OPERATOR IN INTERVENTIONAL DISTRIBUTIONS WITH DECAFLOW

The sampling process consists of sampling first from the prior distribution of the latent variables and from the distribution of the exogenous variables. Then, one can use the generative network ($T_\theta$) to take samples of the rest of variables, changing the components of $\mathbf{u}$ associated with t. Note that $\mathbf{z}$ is not the input of the normalizing flow, but a condition (or *context*). Therefore, $\mathbf{z}$ is transformed neither in the forward nor the reverse pass of the flow.

---

**Algorithm 4** Algorithm to sample from the interventional distribution, $P(\mathbf{x} \mid \text{do}(\text{t} = \alpha))$ with DeCaFlow.

---

1: **function** SAMPLEINTERVENEDDIST($\text{t}, \alpha$)
2:     $\mathbf{z} \sim P_\mathbf{z}$                ▷ Sample a value from the prior of $\mathbf{z}$.
3:     $\mathbf{u} \sim P_\mathbf{u}$            ▷ Sample a value from the observational distribution.
4:     $\mathbf{x} \leftarrow T_{\boldsymbol{\theta},\mathbf{z}}^{-1}(\mathbf{u})$
5:     $\text{t} \leftarrow \alpha$                ▷ Set t to the intervened value $\alpha$.
6:     $\mathbf{u}_\text{t} \leftarrow T_{\boldsymbol{\theta},\mathbf{z}}(\mathbf{x})_\text{t}$        ▷ Change the component of $\mathbf{u}$ associated with t.
7:     $\mathbf{x} \leftarrow T_{\boldsymbol{\theta},\mathbf{z}}^{-1}(\mathbf{u})$
8:     **return** $\mathbf{x}$               ▷ Return the intervened sample.
9: **end function**

---

Additionally, the process to compute the average treatment effect (ATE) involves generating interventional distributions. For example, to compute the ATE comparing two interventions ($\alpha_1, \alpha_2$) in the variable t, we would generate samples of the interventional distributions, $p(\mathbf{x} \mid \text{do}(\text{t} = \alpha_1)), p(\mathbf{x} \mid \text{do}(\text{t} = \alpha_1))$, respectively, and approximate their expectations with MonteCarlo.

$$ATE = \mathbb{E}[\mathbf{x} \mid \text{do}(\text{t} = \alpha_2)] - \mathbb{E}[\mathbf{x} \mid \text{do}(\text{t} = \alpha_1)] \tag{61}$$

Unfortunately, if we were interested in evaluating the ATE on only one variable, y, the process would involve to generate samples of the whole interventional distribution and select only the samples of the interested variable.

## D.3   DO-OPERATOR IN COUNTERFACTUALS WITH DECAFLOW

As part of the abduction step, our model estimates the posterior distribution of hidden confounders given a factual datapoint, $q_\phi(\mathbf{z} \mid \mathbf{x}^{\text{f}})$. Therefore, we can sample from the inferred posterior of the hidden confounders, and use those samples as the condition of the conditional normalizing flows.

---

**Algorithm 5** Algorithm to sample from the counterfactual distribution, $P(\mathbf{x} \mid \text{do}(\text{t} = \alpha))$ with DeCaFlow.

---

1: **function** GETCOUNTERFACTUAL($\mathbf{x}^{\text{f}}, \text{t}, \alpha$)
2:     $q_\phi(\mathbf{z} \mid \mathbf{x}^{\text{f}}) \leftarrow$ Deconfounding network($\mathbf{x}^{\text{f}}$)         ▷ **Abduction:** Get $\mathbf{z}$ from the factual sample.
3:     $\mathbf{z} \sim q_\phi(\mathbf{z} \mid \mathbf{x}^{\text{f}})$         ▷**Abduction**: Sample the posterior distribution.
4:     $\mathbf{u} \leftarrow T_{\boldsymbol{\theta},\mathbf{z}}(\mathbf{x}^{\text{f}})$         ▷**Abduction:** Get $\mathbf{u}$ from the factual sample.
5:     $\text{t}^{\text{f}} \leftarrow \alpha$           ▷**Action:** Set t to the intervened value $\alpha$.
6:     $\mathbf{u}_\text{t} \leftarrow T_{\boldsymbol{\theta},\mathbf{z}}(\mathbf{x}^{\text{f}})_\text{t}$      ▷**Action:** Change the component of $\mathbf{u}$ associated with t.
7:     $\mathbf{x}^{\text{cf}} \leftarrow T_{\boldsymbol{\theta},\mathbf{z}}^{-1}(\mathbf{u})$         ▷ **Prediction:** compute the counterfactual
8:     **return** $\mathbf{x}^{\text{cf}}$             ▷ Return the counterfactual value.
9: **end function**

# E  ADDITIONAL DETAILS ON RELATED WORK OF CAUSAL INFERENCE WITH HIDDEN CONFOUNDERS

In this section, we go deeper into the methods of causal inference in scenarios where there are unobserved confounders.

## E.1  METHODS TAILORED TO GRAPH AND QUERY

First of all, we want to remark that all the following methods have been designed to address causal inferences in specific causal graphs (or subgraphs), therefore they can be used when there exists the causal relationships presented in Fig. 26.

In the following text, we assume the notation introduced in §2, where z is the hidden confounder, t is the intervened variable or treatment and y is the outcome, i.e. the variable where we want to evaluate the causal effects.

We have classified the different approaches depending on the graph that they are designed to address. However, there are two considerations that are common for all these approaches.

First, the methods follow a two-stage process: **i)** extracting a substitute for the unobserved confounder, $\tilde{z}$, using the variables affected by the confounder or instrumental variables, and **ii)** estimating the outcome given this substitute, $\tilde{y} \sim p(y \mid \tilde{z}, t)$. In larger causal graphs, one predictor must be trained for each outcome, and one extractor must be trained per independent confounder.

Second, none of these methods shows the ability of identify *counterfactuals*, since they do not model exogenous variables.

**Presence of null proxies independent of t (Fig. 26a).**   We say **n** to be a null proxy of **z** if it is a child of **z** independent of the outcome, y, given **z**: $n \perp\!\!\!\perp y \mid z$. Methods for estimating causal effects were developed when null proxies of the confounder were available and those proxies are independent of the intervened variable: $n \perp\!\!\!\perp t \mid z$. We can use these proxies to infer a substitute. Among these, Allman et al. [2], Kuroki and Pearl [35] studies the case in which the confounder is categorical and uses matrix factorization to extract a substitute when there are at least three Gaussian proxies [2], when the conditional distribution of the confounder given the proxy is known or when other proxies are available [35]. Kallus et al. [26] also employ matrix factorization to cases where the confounder is continuous and the relation with the covariates and the treatment (but not with the outcome) is linear. In addition, Kallus et al. [27] uses kernel functions to extract the substitute confounder when the generators are nonlinear. The most relevant method based on deep generative methods is proposed by Louizos et al. [38], consisting of a VAE to extract the substitute confounder when several null proxies are available, although there is no theoretical guarantee of its operation  and has been shown to struggle with complex distributions in practice [52]. Finally, Miao et al. [42] offers a regression-based approach to estimate the unobserved confounder under *equivalence*, which assumes that any model of the joint achieves element-wise transformations of the latents, which is not feasible to check: $\tilde{p}(t, z \mid n) = p(t, V(z) \mid n)$. The graph in which all these methods operate can be found in Fig. 26a.

**Presence of two proxies: null and not null (Fig. 26b).**   When the null proxies affect treatment (see Fig. 26b: the proxy, **n**, affects treatment t), Miao et al. [41] offers theoretic guarantees of causal identifiability in the presence of another proxy, **w**, and completeness conditions. The proxy **w** can be active, that is, it can directly affect y. Practically, in Tchetgen et al. [61] the two-stage proximal least squares (P2SLS) we can find the method to infer the substitute confounder from $p(w \mid t, n)$. P2SLS can be implemented using neural networks to achieve greater flexibility.

After the publication of Miao et al. [41], several follow-up works have emerged that aimed to estimate the bridge function, solving Eq. 12 explicitly. For example, Cui et al. [11] designed a doubly-robust estimator of the ATE by estimating the bridge function semiparametrically, and Kompa et al. [33], Mastouri et al. [40] apply moment restrictions to estimate the bridge function using deep neural networks. Other works propose multiple-robust methods when confounder are categorical [58]

**Instrumental variable (Fig. 26c).**   Another condition that allows causal inference is the presence of instrumental variables (IVs), i.e. variables that affect only the treatment and are independent of both the unobserved confounder and the outcome given the treatment (in Fig. 26c, **n** is an IV). In linear DGP, Angrist and Pischke [4], Pearl [47] demonstrates that a two-stage regression process mitigates the confounding bias as the only effect that flows from the instrumental variable to the outcome is through treatment. A substitute of the confounder is extracted by computing the conditional distribution of the treatment given the instrumental variable: $\tilde{z} \sim p(t \mid n)$. Furthermore, [20] develops an extension of this theory to include arbitrarily complex nonlinear DGP, designing a two-step deep approach, based on neural networks.

**Multitreatment affected by a common confounder (Fig. 26d).**   Finally, the multitreatment scenario (Fig. 26d) has been studied by Ranganath and Perotte [51], Wang and Blei [65]. It is called multitreatment because all covariates can be seen as a treatment over the outcome, y. It is assumed that, in the true DGP, there exist several covariates that are independent given the unobserved confounder. Therefore Wang and Blei [65] propose to use a factorization model, such as probabilistic PCA or Poisson Matrix Factorization, to infer the substitute confounder. A factorization model assumes that the distribution of all

the treatments factorizes in the following way: $p(\mathbf{t}, \mathbf{z}) = p(\mathbf{z}) \prod_{i=1}^{d} p(\mathrm{t}_i \mid \mathbf{z})$, which should allow to construct a substitute of the confounder from the posterior of $\mathbf{z}$: $\tilde{\mathbf{z}} \sim \tilde{p}(\tilde{\mathbf{z}} \mid \mathbf{t})$. However, D'Amour [12] provide counterexamples showing that the Deconfounder does not achieve nonparametric identification without additional assumptions. Notably, one of the alternatives D'Amour [12] highlights is the use of proxy variables—precisely the approach adopted by DeCaFlow.

On the other hand, similar to Wang and Blei [65], Ranganath and Perotte [51] proposes a method that uses a VAE as a factorization model, adding a regularization term to reduce the additional mutual information between the estimated confounder and treatment $\mathrm{t}_j$ given the rest of treatments $\mathbf{t}_{-j}$. However, the theoretical guarantees of this approach require an infinite number of treatments to achieve unbiased estimates of causal effects.

Wang and Blei [66] connect the ideas of Miao et al. [41] and Wang and Blei [65] ensuring causal identification in the multitreatment setting when it is known that some of the treatments *can act as null proxies*, that is, they do not affect the outcome. This assumption allows them to provide theoretical guarantees when the number of treatments does not tend to be infinite. Even so, a factorization model such as the one Wang et al. [67] propose can only model independent treatments, given the hidden confounder, which greatly limits its usefulness.

**How is Deconfounder Wang and Blei [65, 66] related to our work.** As DeCaFlow does, Deconfounder infers the posterior distribution of the substitute of the confounder from the observational data using a generative model. However, the application of a factorization model restricts the structural dependencies that we can model. For example, the Deconfounder cannot model the structural dependencies of Fig. 26b, since the factorization model assumes $\mathbf{n} \perp\!\!\!\perp \mathrm{t} \perp\!\!\!\perp \mathbf{w} \mid \mathbf{z}$. In contrast, the DeCaFlow uses a causal flow, which does allow this dependencies because the causal graph is encoded in the flow.

We also stress that DeCaFlow models the whole confounded SCM, including the exogenous variables. This allows to compute *counterfactuals* and train in a query-agnostic manner. In contrast, Deconfounder cannot compute counterfactuals and needs of a separate model per query.

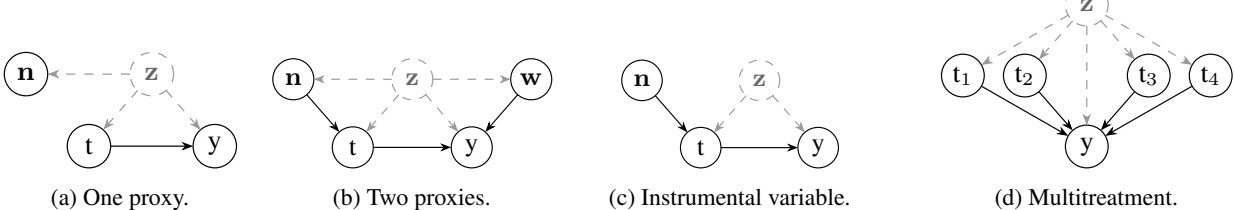

(a) One proxy.   (b) Two proxies.   (c) Instrumental variable.   (d) Multitreatment.

Figure 26: *Ad-hoc* graphs. **(a)** Allman et al. [2], Kallus et al. [26, 27], Kuroki and Pearl [35], Louizos et al. [38], Miao et al. [42] address the case where $\mathbf{n}$ is independent of t. **(b)** Miao et al. [41] is designed for the case where there exist two proxies. **(c)** Graph with an instrumental variable, but this graph is out of the scope of our framework. **(d)** Ranganath and Perotte [51], Wang and Blei [65, 66] are designed for the multitreatment setting.

## E.2 CGM WITH UNOBSERVED CONFOUNDERS

There exist several works that employ causal generative models (CGMs) in the presence of hidden confounders. We explain here the differences with our proposal, highlighting the practical advantages of DeCaFlow.

**Neural Causal Models (NCMs)**   Xia et al. [71] proposed a class of sequential causal generative models where each structural equation—i.e., the functional relationship between a variable and its parents in the causal graph—is modeled by a distinct neural network. The model is trained end-to-end to jointly learn all structural mechanisms. Beyond estimation, NCMs aim to determine whether a given causal query is identifiable from the data-generating process.

To assess identifiability, their method trains two versions of the model: one that maximizes and one that minimizes the likelihood of the query under consideration. If both yield the same outcome, the query is deemed identifiable. This approach formalizes identifiability as an empirical condition based on optimization agreement.

However, the framework has significant practical constraints: **i)** it only supports finite discrete variables, typically binary and low-dimensional, due to tractability constraints; **ii)** it assumes that the true observational distribution is available for training; **iii)** two models must be trained per query, leading to high computational cost; and **iv)** identifiability status is only revealed post-training, offering no guidance before model execution.

To address counterfactual reasoning, Xia et al. [72] extended NCMs to estimate queries involving latent exogenous variables. However, their approach relies on rejection sampling to infer hidden confounders, which is inefficient and unsuitable for continuous or high-dimensional settings, thus limiting its applicability in real-world scenarios.

In contrast, our approach addresses these limitations. First, we provide a principled criterion to estimate the identifiability of a query *prior* to model training. Second, our framework supports continuous variables and scales to high-dimensional settings. Third, we train a single model that jointly estimates all causal mechanisms and enables efficient inference of counterfactuals. Fourth, we use variational inference to approximate the posterior of hidden confounders, avoiding the inefficiency of rejection-based methods. Finally, we guarantee the identifiability of exogenous variables (in the sense of Xi and Bloem-Reddy [70]) by leveraging the theoretical framework of the causal flows [25]. As a result, our method is substantially more efficient and suited to real-world applications.

**Modular Causal Generative Models**    Rahman and Kocaoglu [50] introduce a modular framework for high-dimensional causal inference, where variables influenced by the same hidden confounder are modeled jointly in end-to-end submodules. A key advantage of this approach is the ability to incorporate pretrained models into submodules, enabling flexible modeling of complex or structured variables when the modular criterion holds. The method supports continuous and discrete variables and uses adversarial training to match observational distributions. Symbolic identifiability is computed using the algorithm of Jaber et al. [24], and they prove that identifiable queries remain estimable under their modular decomposition. However, the framework does not support counterfactual inference and proximal learning and is based on adversarial optimization.

Compared with this method, our approach trains a single end-to-end model, estimates both observational and counterfactual distributions also in proximal settings, achieves identifiability in the exogenous distributions, and enables efficient inference with broad applicability to real-world settings.

**Counterfactual Identifiability of Bijective Causal Models**    Nasr-Esfahany et al. [43] propose a sequential causal model using conditional normalizing flows to map exogenous to endogenous variables. The model focuses on counterfactual inference under backdoor and instrumental variable (IV) settings, with identifiability proven only for discrete cases. Proxy variables are not considered, and the use of invertible mappings over discrete domains makes theoretical claims less robust. Although the model claims support for continuous data, guarantees are restricted to discrete IV scenarios. It does not model observational or interventional distributions and lacks parameter amortization due to its sequential structure.

In contrast, our method supports continuous variables, models both observational and interventional distributions, and enables counterfactual inference under general confounding and proxy settings. It uses a single end-to-end model and scales efficiently to real-world data.

**Learning Functional Causal Models with Generative Neural Networks**    Goudet et al. [18] propose a method for causal discovery rather than causal inference under unobserved confounding. Given a Markov equivalence class or graph skeleton, their approach uses generative neural networks to model each causal direction and selects the graph that best matches the observational distribution, evaluated via maximum mean discrepancy (MMD). The model is trained sequentially and assumes no hidden confounders. While not directly comparable to our work, such causal discovery tools may serve as a preprocessing step when the causal graph is unknown, enabling downstream application of models—such as ours—that assume a known and correct structure.

# F    ALGORITHMS FOR CAUSAL QUERY IDENTIFICATION

As explained in §4.2, we can ask DeCaFlow to solve any causal query, but we do not have the guarantee that the estimation that DeCaFlow returns is correct unless the query is identifiable. Therefore, we provide the practitioner with algorithms to check the identifiability of causal queries.

**Specific treatment-outcome pair.**    We start presenting the Alg. 6 to identify a causal query specifying the pair treatment and outcome, which is valid for estimating the interventional distribution of the outcome, $p(\mathrm{y}\,|\,\mathrm{do}(\mathrm{t}), \mathbf{c})$, and the counterfactual one, $p(\mathrm{y^{cf}}\,|\,\mathrm{do}(\mathrm{t}), \mathbf{x^f})$, since we postulated in §4 that the latter is identifiable if the former is.

We have employed this algorithm in all the paths of Sachs and Ecoli70 datasets to check the identifiability of all the direct causal effects, where y is a child of t, in order to get a visual representation of the identifiable queries of a complex graph. However, due to the large number of possible causal queries resulting from all edge combinations in the 43-node Ecoli70 dataset, we have not analyzed identifiability for all undirected queries. If one is interested in evaluating a query which involves several outcomes, $\{\mathrm{y}_1, \mathrm{y}_2, ..., \mathrm{y}_O\}$, one causal query per $\mathrm{y}_i$ should be evaluated.

**Evaluation on all the variables.**    Although the Alg. 7 consist of applying Alg. 6 iteratively, we also find it interesting to include the extension to identify causal queries evaluated on all variables in the dataset, which is useful for using DeCaFlow as a generative model for the interventional distribution, $p(\mathbf{x} \mid \mathrm{do}(\mathrm{t}))$, or offering complete counterfactual samples, $p(\mathbf{x^{cf}} \mid \mathrm{do}(\mathrm{t}), \mathbf{x^f})$, intervening in a specific variable, $\mathrm{t} \subset \mathbf{x}$.

## F.1    PIPELINE FOR USING DECAFLOW

**Algorithm 6** Identification of causal queries that include intervention and outcome (t, y)

---

**Require:** Graph $\mathcal{G}$, intervention variable t, outcome variable y, covariates $\mathbf{c}$, hidden variables $\mathbf{z}$
**Ensure:** Boolean indicating if query is identifiable
1: $\mathbf{z} \leftarrow$ hidden variables that are parents of both t and y
2: **return** True **if** $\mathbf{z}$ is $\emptyset$                                        ▷ Unconfounded is identifiable
3: **for all** $\mathbf{z}_k \in \mathbf{z}$ **do**
4:     **Comment:** Each $\mathbf{z}_k$ is an independent component of $\mathbf{z}$
5:     n-proxies $\leftarrow$ children of $\mathbf{z}_k$ $d$-separated from t given $(\mathbf{z}, \mathbf{c})$
6:     w-proxies $\leftarrow$ children of $\mathbf{z}_k$ $d$-separated from y given $(\mathbf{z}, \mathbf{c})$
7:     **if** there exist $\mathbf{n} \in$ n-proxies and $\mathbf{w} \in$ w-proxies such that $\mathbf{n}$ is $d$-separated from $\mathbf{w}$ given $(\mathbf{z}, \mathbf{c})$ **then**
8:         $\mathbf{z}_k$ is deconfounded
9:     **end if**
10: **end for**
11: **return** all $\mathbf{z}_k$ are deconfounded

---

**Algorithm 7** Identification of causal queries, intervening in t and evaluating in all variables

---

**Require:** Graph $\mathcal{G}$, intervention variable t, hidden variables $\mathbf{z}$
**Ensure:** Boolean indicating if the interventional distribution is identifiable
1: $\mathbf{z} \leftarrow$ hidden variables that are parents of t
2: **for all** $\mathbf{x}_i \in$ descendants of t **do**
3:     **Comment:** Evaluate only on descendants of the intervention
4:     Check $(\mathrm{t}, \mathbf{x}_i)$ identifiability with Alg. 6
5: **end for**
6: **return** all $(\mathrm{t}, \mathbf{x}_i)$ are identifiable

---

Our framework provides a systematic approach to solving causal queries by integrating DeCaFlow, a model trained on observational data, with algorithms designed for query identifiability analysis.

As depicted in the pipeline, the framework takes as input a dataset $\mathcal{D}$, a causal graph $\mathcal{G}$, and a set of $N$ interesting queries $\{Q_i\}_{i=1}^N$. The process begins by training DeCaFlow on $\mathcal{D}$ and $\mathcal{G}$, enabling it to learn the confounded SCM, $\mathcal{M}$.

Simultaneously, the identifiability of each causal query $Q_i$ is assessed using dedicated algorithms (Alg. 6 and Alg. 7). If $Q_i$ is identifiable, the trained DeCaFlow is used to estimate $Q_i(\mathcal{M})$ (Alg. 4 and Alg. 5), yielding the estimated causal effect $\hat{Q}_i(\mathcal{M})$. If $Q_i$ is not identifiable, the framework indicates that answering the query is not feasible given the available data and causal structure. Other causal queries can be answered by the model without retraining, provided that their identifiability is verified beforehand.

This workflow ensures a principled approach to causal inference, leveraging both data-driven modelling and theoretical guarantees on identifiability. Both the DeCaFlow model and the algorithms for query identifiability and estimation will be included in the code that we will provide upon acceptance.

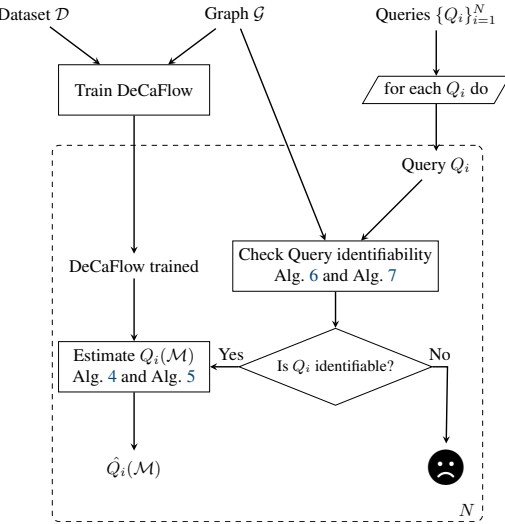

Figure 27: **Block diagram of the pipeline.**

**Validation with interventional data.** As a final step in the pipeline in real-world scenarios, especially in sensitive applications, we encourage practitioners to validate the framework with interventional data. Causal queries such as *average treatment effects* (ATEs) can be validated if a randomized experiment is available in which interventions are carried out on the treatment variable.

However, in cases where experiments on the required variable are not available, our framework can still be partially validated by assessing the completeness of the inferred hidden confounder given the observed proxies. This can be done by evaluating causal effects in another causal query that shares the same hidden confounder. Specifically, if a causal query $Q_1$

lacks interventional data, but another query $Q_2$ involving the same hidden confounder is estimated correctly, the inferred confounder of $Q_2$ can be postulated as a valid substitute for estimating $Q_1$. This indirect validation method provides a way to assess the reliability of our framework without requiring direct interventions for every confounded query.

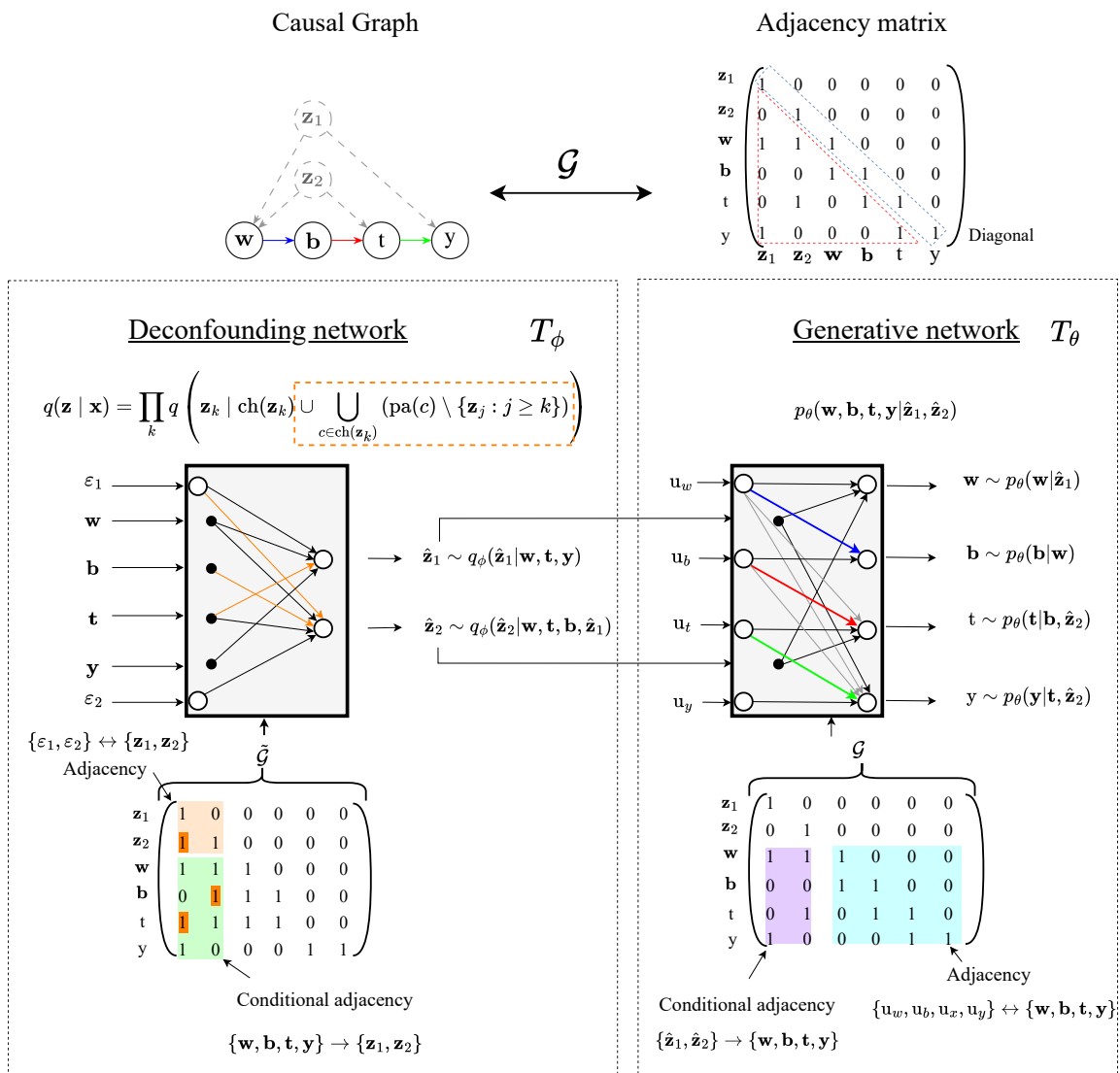

Figure 23: Example of complete DeCaFlow architecture, applied to the specific graph of Fig. 9. Both the deconfounding network and the generative network are conditional normalizing flows that factorize the distributions of the posterior and endogenous variables following Eq. 3 and Eq. 2, respectively. Within networks, functional dependencies are represented following the compacted version of Javaloy et al. [25, Fig. 4(c)]. The orange edges of the encoder corresponds to the collider association in the posterior factorization, and $\tilde{\mathcal{G}}$ encodes that associations.

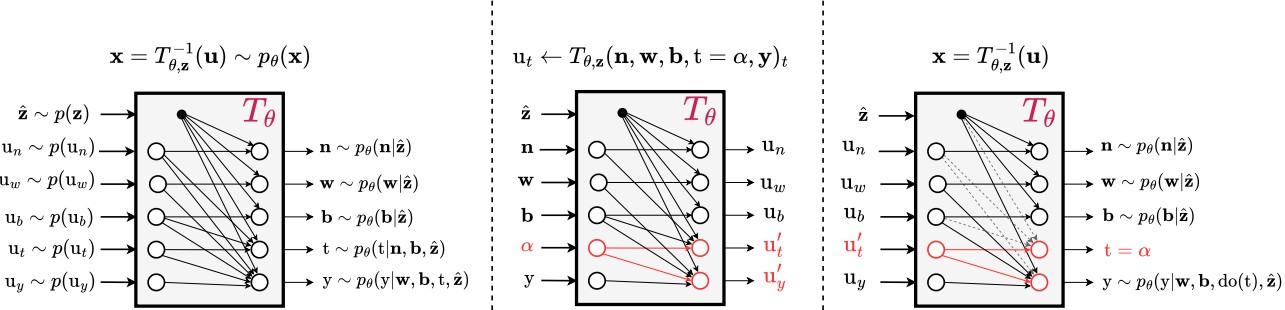

Figure 24: Schematic view of sampling process from interventional distribution in graph of Fig. 6, intervening in t. By sampling from the prior of the hidden confounders, $p(\mathbf{z})$, and the base distribution of the exogenous variables, $p(\mathbf{u})$, we get samples of the empirical marginal interventional distribution $p_\theta(\mathbf{y} \mid \mathrm{do}(t))$ through MonteCarlo integration. Note that sampling from the interventional distribution only requires the generative network, $T_\theta$. Dashed gray arrows represent the cancellation of causal effect due to the intervention.

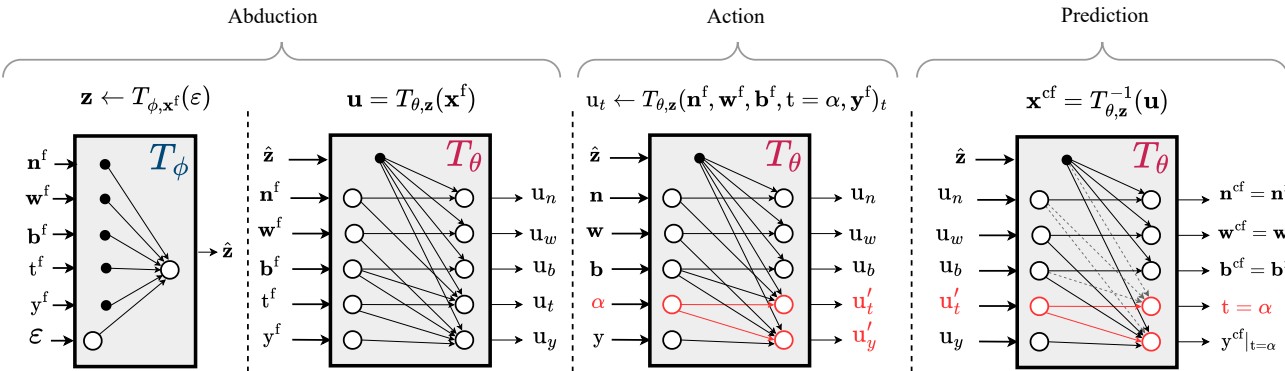

Figure 25: Schematic view of counterfactual inference with the graph of Fig. 6, intervening in t. This inference can be done from a single point, we only sample from $\varepsilon$. Both thedeconfounding network, $T_\phi$, and the generative network, $T_\theta$, are needed. Dashed gray arrows represent the cancellation of causal effect due to the intervention.