# OpenReview forum: "DeCaFlow: A Deconfounding Causal Generative Model"
_auai.org/UAI/2025/Workshop/TPM — TPM 2025_

### Official Review · Reviewer_oo3s · 2025-06-12
**(Causal) NFs to extend the class of identifiable queries**

**Rating:** 3

**Review:**

The paper presents a novel approach to causal analysis based on normalising flows (NFs). Given a causal graph over continuous variables and an observational dataset, the algorithm might estimate identifiable interventional and counterfactual queries. The authors start from an existing causal version of NFs, which is extended to the confounded case here. An ELBO task is eventually required for the model training.

Interestingly, the authors show that their approach goes beyond standard identifiability, as it can be done by Pearl's do calculus. In this sense, they can estimate queries under an extended notion of identifiability. This is a clear advance, and the paper deserves acceptance. After more extensive validation and more details are added, the paper has a reasonable chance of being accepted at a major conference.

**Nominate For Best Paper:**

["Yes"]

---

### Official Review · Reviewer_fV6D · 2025-06-16
**New causal generative model under hidden confounding**

**Rating:** 3

**Review:**

This paper introduces a novel causal generative model, DeCaFlow, capable of answering interventional and counterfactual queries. The model consists of an encoder-decoder architecture where the encoder predicts the hidden confounders given observed variables, and the decoder follows a (conditional) causal normalizing flow mapping from exogeneous to endogenous variables.

Overall, the paper is an exciting and significant new approach to causal inference due to DeCaFlow's ability to perform inference for all marginal interventional/counterfactual queries once trained on a given causal graph and dataset. The key insight on parameterizing a confounded SCM using conditional flows is very interesting, and the methodological choices on e.g. the encoder and empirical results are thorough. The manuscript is also very well put together and readable considering the space constraints. As such I strongly recommend acceptance.

Comments/Questions/Suggestions:
- Can we still employ DeCaFlow given partial knowledge of the causal graph?
- It wasn't clear to me if Prop 4.1 (on proximal identifiability) is essential or specific to the DeCaFlow method, or a separate identifiability result (which DeCaFlow can exploit).
- It might be worth elaborating on which interventional/counterfactual query sets can or can't be estimated tractably using this approach (and why): e.g. can one compute ATEs with interventions on multiple variables, or counterfactual queries with partial observations? Does this depend on the choice of the encoder?
- Some other related works on causal generative modeling under hidden confounding: [1] uses probabilistic circuits to tractably compute interventional queries identifiable by do-calculus; and [2] proposes sampling using conditional generative models following the ID algorithm.
[1] Wang & Kwiatkowska (2023). Compositional Probabilistic and Causal Inference using Tractable Circuit Models. AISTATS 2023.
[2] Rahman et al. (2024). Conditional Generative Models are Sufficient to Sample from Any Causal Effect Estimand. NeurIPS 2024.

**Nominate For Best Paper:**

["Yes"]